



# A derivative-free optimisation method for global ocean biogeochemical models

Sophy Oliver[1], Coralia Cartis[2], Iris Kriest[3], Simon Tett[4], and Samar Khatiwala[1]

[1]Department of Earth Sciences, University of Oxford, South Parks Road, Oxford OX1 3AN, UK
[2]Mathematical Institute, University of Oxford, Radcliffe Observatory Quarter, Woodstock Road, Oxford, OX2 6GG, UK
[3]GEOMAR Helmholtz-Zentrum für Ozeanforschung Kiel, Düsternbrooker Weg 20, 24105 Kiel, Germany
[4]School of GeoSciences, University of Edinburgh, Edinburgh, UK

**Correspondence:** Sophy Oliver (sophy.oliver@oriel.ox.ac.uk)

**Abstract.** The performance of global ocean biogeochemical models, and the Earth System Models in which they are embedded, can be improved by systematic calibration of the parameter values against observations. However, such tuning is seldom undertaken as these models are computationally very expensive. Here we investigate the performance of DFO-LS, a local, derivative-free optimisation algorithm which has been designed for computationally expensive models with irregular model-data misfit landscapes typical of biogeochemical models. We use DFO-LS to calibrate six parameters of a relatively complex global ocean biogeochemical model (MOPS) against synthetic dissolved oxygen, inorganic phosphate and inorganic nitrate "observations" from a reference run of the same model with a known parameter configuration. The performance of DFO-LS is compared with that of CMA-ES, another derivative-free algorithm that was applied in a previous study to the same model in one of the first successful attempts at calibrating a global model of this complexity. We find that DFO-LS successfully recovers 5 of the 6 parameters in approximately 40 evaluations of the misfit function (each one requiring a 3000 year run of MOPS to equilibrium), while CMA-ES needs over 1200 evaluations. Moreover, DFO-LS reached a "baseline" misfit, defined by observational noise, in just 11–14 evaluations, whereas CMA-ES required approximately 340 evaluations. We also find that the performance of DFO-LS is not significantly affected by observational sparsity, however fewer parameters were successfully optimised in the presence of observational uncertainty. The results presented here suggest that DFO-LS is sufficiently inexpensive and robust to apply to the calibration of complex, global ocean biogeochemical models.

## 1 Introduction

Ocean biogeochemical models are a key tool in understanding the cycling of nutrients and carbon in the ocean. They are used to quantify the uptake of greenhouse gases such as $CO_2$ emitted by human activity, of which the ocean has absorbed roughly a third since the start of the industrial revolution (Khatiwala et al., 2009; DeVries, 2014), as well as assess the impact of increasing concentrations of greenhouse gases on ocean ecosystems. Such models are also an important component of the Earth System Models (ESMs) used to project future climate change. In global ocean biogeochemical models the complex interactions between biota, nutrients, oxygen and carbon are typically heavily parameterized. The performance of such models can be improved by either subjective manual or systematic tuning of the parameter values against observations. The latter



uses numerical optimisation algorithms which seek to find the minima of a "misfit function"–often defined as the root mean

squared difference between the model and observations–within the parameter space. However, biogeochemical models are seldom subjected to such tuning because of their large computational expense and the long spin-up time required for chemical and biological tracers to reach equilibrium (Wunsch and Heimbach, 2008; Khatiwala et al., 2012). Moreover, optimisation algorithms must be able to navigate a generally irregular misfit landscape. Efficient and robust optimisation methods are thus of considerable interest to the ocean biogeochemical and broader climate modeling community.

Previous ocean biogeochemical calibration studies have more frequently been carried out on computationally less expensive 0-dimensional (e.g. Kidston et al., 2013) and 1-dimensional models (e.g. Chen and Smith, 2018; Xiao and Friedrichs, 2014; Ward et al., 2010; Spitz et al., 1998), regional models (e.g. Melbourne-Thomas et al., 2015; Zhao et al., 2005), or steady-state global models (e.g. Kwon and Primeau, 2006, 2008). However, with the aid of fast "offline" circulation schemes, such as the Transport Matrix Method (Khatiwala et al., 2005; Li and Primeau, 2008) which can be applied to time-dependent

biogeochemical models, more recently, complex global ocean biogeochemical models have also begun to be systematically optimised to observations (e.g. Kriest et al., 2017, 2020; Sauerland et al., 2019; Niemeyer et al., 2019; Kriest, 2017).

Optimisation methods can generally be split into two broad categories. Derivative-based algorithms such as Gauss-Newton (Hartley, 1961) use finite differences or adjoints to calculate derivatives within the parameter space to locate minima. They can be both computationally expensive and generally less robust on or even unsuitable for noisy problems. Derivative-free

algorithms (Conn et al., 2009) in contrast can be less computationally expensive and are better adapted to handle noisy misfit functions. An example of the latter is "Covariance Matrix Adaptation Evolution Strategy" (CMA-ES; Hansen, 2016). CMA-ES was applied by Kriest et al. (2017) to optimise six parameters within the Model of Oceanic Pelagic Stoichiometry (MOPS; Kriest and Oschlies, 2015), by minimising a globally averaged misfit incorporating annual mean dissolved inorganic phosphate, nitrogen and oxygen. This constituted one of the first successful attempts at systematic tuning of a relatively complex global

biogeochemical model. CMA-ES was subsequently used by Sauerland et al. (2019) for multiobjective calibration of MOPS by including oxygen minimum zones as a misfit metric, and by Kriest et al. (2020) who compared the influence of different general circulation models on parameter optimisation.

While the development and application of CMA-ES is an important step forward, it is still computationally too expensive for routine use. In Kriest et al. (2017), for example, the misfit function had to be evaluated at least 950 times to achieve a sufficiently

low misfit. As each evaluation requires running the biogeochemical model to equilibrium (3000 years in that study), this would be prohibitively expensive for the more complex models run at resolutions typical of the current generation of ESMs. Here, we explore the application of another, computationally less expensive algorithm, "Derivative Free Optimisation by Least Squares" (DFO-LS; Cartis et al., 2019), to the same problem set-up as in Kriest et al. (2017). We first compare the performance of CMA-ES and DFO-LS to optimise six biogeochemical parameters against the output of a reference run of MOPS where the

parameters are known. We examine in this "twin" experiment the ability of the algorithms to recover the true parameters, and the computational cost incurred. True oceanic observations contain observational uncertainty, therefore we also investigate the impact of optimising in the presence of observational uncertainty by adding noise to the synthetic observations. Lastly, we evaluate the performance of DFO-LS when given sparse data. Sparse scattered oceanic observations are commonly interpolated





onto a regular grid, introducing significant error, especially in regions such as the Southern Ocean with poor data coverage.
The structure of the paper is as follows: Section 2 describes the methodology, Section 3 the results, 4 the discussion and 5 the conclusions.

## 2  Methodology

### 2.1  Ocean biogeochemical model

The Model of Oceanic Pelagic Stoichiometry (MOPS-2.0) is a global ocean biogeochemical model, which simulates the cy-
cling of 9 biogeochemical tracers, namely dissolved inorganic and organic phosphate, dissolved inorganic nitrate, dissolved oxygen, phytoplankton, zooplankton, detritus, dissolved inorganic carbon and alkalinity (Kriest and Oschlies, 2013, 2015). MOPS is coupled to the Transport Matrix Method (TMM; Khatiwala et al., 2005; Khatiwala, 2007, 2018), an efficient numerical method for "offline" simulation of biogeochemical tracers. In this study we use monthly mean transport matrices and other physical forcing fields (including temperature, salinity, sea ice and winds) derived from a relatively coarse resolu-
tion ($2.8° \times 2.8° \times 15$ levels) configuration of MITgcm (Marshall et al., 1997) driven by climatological momentum, heat and freshwater fluxes (Dutkiewicz et al., 2005).

### 2.2  Biogeochemical model parameters

The behaviour of MOPS is controlled by several parameters, of which we have chosen 6 to consider for calibration, based on the previous optimisation study by Kriest et al. (2017). The detailed definitions and possible ranges of these parameters are
described in that paper. Briefly, 4 of these parameters are mainly restricted to the epipelagic and mesopelagic zones of the ocean, as they involve phytoplankton and zooplankton. $I_\mathrm{C}$ and $K_\mathrm{PHY}$ are the phytoplankton half-saturation constants for light absorption and phosphate uptake, respectively. $\mu_\mathrm{ZOO}$ is the zooplankton maximum grazing rate and $k_\mathrm{ZOO}$ the zooplankton quadratic mortality rate. The remaining two parameters influence the remineralisation and sinking of particulate organic matter (POM). $R_\mathrm{O_2:P}$ is the ratio of oxygen consumption to phosphate release during remineralisation when oxygen is available, and
$b^*$ is the exponent of the "Martin curve", a power law function that describes the attenuation of POM flux with depth (Martin et al., 1987).

### 2.3  Optimisation Algorithms

Optimisation algorithms iteratively evaluate the misfit between model and observations, then vary the model parameter inputs with the aim of finding a lower misfit. Here, every evaluation of the misfit requires running the biogeochemical model to
equilibrium (3000 years), then calculating the misfit between the model outputs and real (or synthetic) observations of dissolved oxygen, phosphate and nitrate. In general the misfit "landscapes" of biogeochemical models tend to be quite irregular, with many local minima. To determine if an optimiser can find the global minimum within the misfit function landscape "twin" experiments are used, whereby the misfit is calculated between the model outputs and synthetic observations. The synthetic





observations are created by the model with a known parameter configuration, therefore the global minimum (and optimal
parameter values) are known. We use twin experiments to compare the performance of two different optimisation algorithms.

### 2.3.1 CMA-ES

The Covariance Matrix Adaptation Evolution Strategy (CMA-ES; Hansen, 2016) is a widely used stochastic evolutionary
algorithm, for use on a "black box" misfit function. By design, CMA-ES is an unconstrained solver, that is, parameters are not
restricted to be within specified bounds. To ensure that parameters lie within reasonable bounds, a penalty score is added to
the misfit when any parameter value goes outside of their specified range. During each iteration CMA-ES evaluates the misfit
function multiple times with various parameter configurations and updates a covariance matrix. It returns only the parameter
configurations which provide the best misfits to a multivariate normal distribution of parameters, then in the next iteration
it randomly draws several more parameter configurations, and repeats. With each iteration the population should be guided
towards areas of the parameter landscape which provide lower expected misfit values, and therefore aim to converge on the
parameter configurations which provide the best misfits. This process has been well illustrated by Kriest et al. (2017, see
their Fig. 2), who previously used CMA-ES to optimise MOPS. CMA-ES carries out a global search of the parameter space,
therefore it seeks to find the minimum over the parameter space. In order to achieve this it requires many function evaluations,
and therefore can be quite computationally expensive in practice. The CMA-ES code used in this study, which is based on
the $(\mu/\mu_{\mathrm{W}},\lambda)$-CMA-ES algorithm of Hansen (2016), was sourced from Kriest et al. (2017). It is summarised in Appendix A.
As in the previous Kriest et al. study, we use a population size of 10, i.e., in each iteration of CMA-ES, the misfit function is
evaluated 10 times.

### 2.4 DFO-LS

Derivative-free optimisation using least squares (DFO-LS) is an iterative algorithm for minimising a function $f(\mathbf{x})$ (Cartis
et al., 2019), where $\mathbf{x} \in \mathbb{R}^n$ is the vector of parameters, each of which is constrained within specified bounds. DFO-LS can
take into account the individual terms in the misfit function and mathematically incorporate their individual function structures
into one single misfit function, which it then minimises. Mathematically, DFO-LS solves the nonlinear least-squares problem:

$$\min_{\mathbf{x} \in \mathbb{R}^n} f(\mathbf{x}) = \sum_{i=1}^{d} r_i(\mathbf{x})^2, \tag{1}$$

where the $r_i(\mathbf{x})$ are individual terms in the misfit function. DFO-LS starts at an initial location within the parameter space,
then moves through the space to provably find a local minimum. The algorithm is illustrated in Fig. 1 and summarised in
Appendix B.

DFO-LS must be given a starting location within the parameter space from which to initialise. In the initial iteration of
DFO-LS, the misfit function is evaluated at the starting location and at $n$ additional locations nearby, for a total of $n+1$
function evaluations. In subsequent iterations only one function evaluation is needed. This is important to note because it




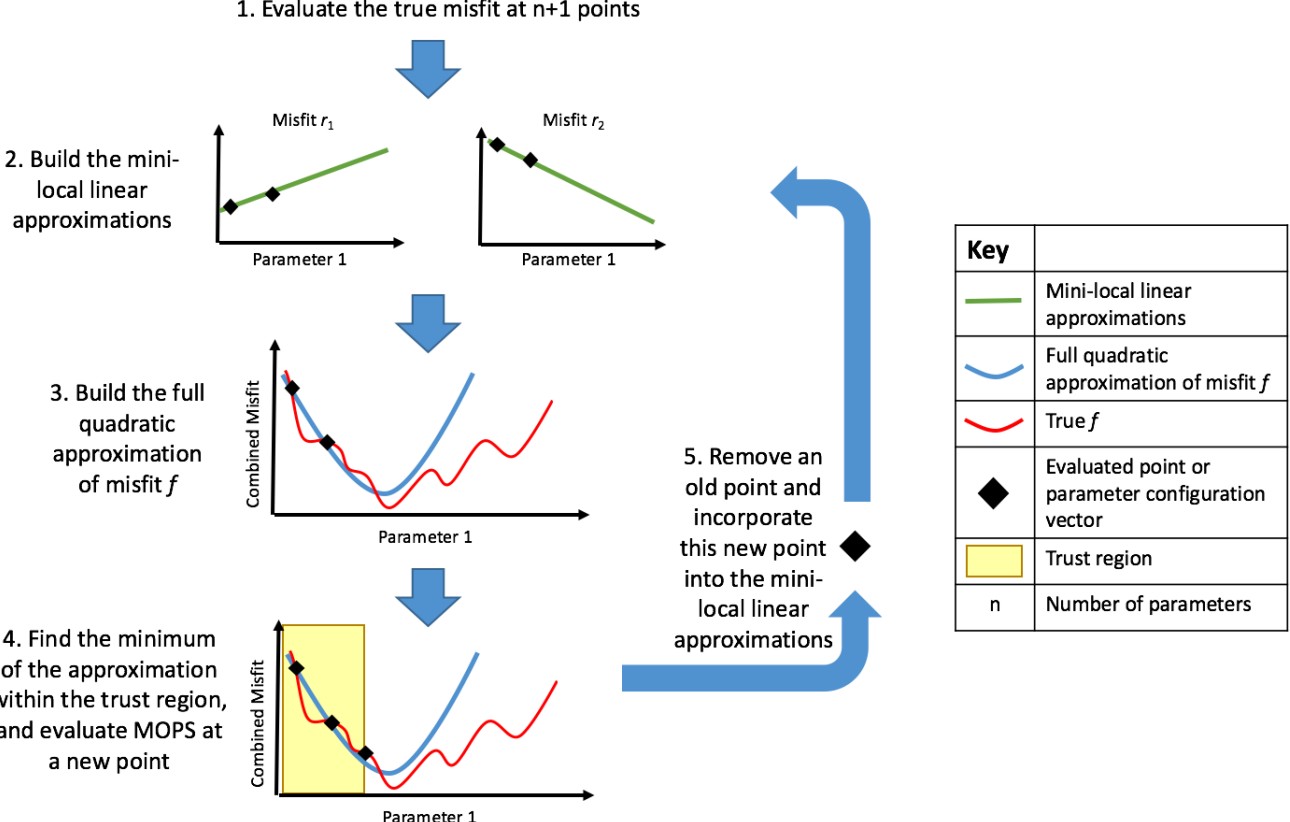

**Figure 1.** Schematic of DFO-LS optimising one parameter. 1) Two individual misfits $r_i$ are evaluated at two locations in parameter space. 2) Two mini-local regressions are built (green lines) using these two $r_i$ points (black diamonds). 3) Information from these are combined to build a quadratic approximation (blue line) to the true misfit function. 4) Within the trust-region (shaded in yellow) the minimum of the approximation is found, at which the true misfit function is evaluated if accepted. 5) This new information is used to update the mini-local regressions. Steps 2-5 are then repeated until the specified termination criterion or maximum evaluations is reached.

means the computational expense of the initial sampling by DFO-LS increases only linearly with the number of parameters to

be optimised.

Using this set of evaluated points, DFO-LS creates a quadratic approximation to the underlying true (unknown) misfit function (Cartis et al., 2019) and calculates the minimum of this function within a "trust region" centred around the starting point. The true misfit function is then evaluated at this location. If it is found to be worse than the misfit at the existing $n+1$ points, it is rejected. The trust region is shrunk and the procedure is repeated. On the other hand, if it is found to be lower than

the best of the $n+1$ points then it is accepted, and the point corresponding to the highest misfit amongst the previous $n+1$ points is discarded. A new quadratic approximation is calculated for this $n+1$ set of points, and the procedure is repeated. Thus, at any iteration DFO-LS keeps track of $n+1$ points in parameter space and the point with the lowest misfit is considered as that iteration's best set of parameters. The algorithm is terminated based on three specified criteria: 1) the maximum allowed


number of function evaluations is exceeded, 2) the trust region radius is shrunk below a specified size, and 3) misfit reduction
progress is identified as being too slow.

Unlike CMA-ES, DFO-LS is more of a "local" optimisation method. However, there is strong numerical evidence from
the derivative-free optimiser Py-BOBYQA, upon which DFO-LS is based (Cartis et al., 2018), that it is able to find global
minima . To increase the likelihood of finding the global minimum, DFO-LS can either be manually re-initiated from different
starting locations in the parameter space, or automatically "restarted" once it determines that the reduction in the misfit is
progressing too slowly. During a restart the trust region expands, allowing DFO-LS to search for points potentially outside the
local minimum it may be trapped in, and move towards a lower minimum elsewhere. This can be done by moving the current
$n + 1$ points in parameter space to geometry-improving points (Cartis et al., 2019) within the new trust region. This is called a
"soft" restart and it encourages DFO-LS to move out of the neighbourhood of a local minimum, hopefully towards the global.
The alternative is a "hard" restart, in which DFO-LS re-evaluates $n + 1$ points within the new trust region. This is much more
computationally expensive than a soft restart, therefore we don't use it here, although soft restarts are allowed. To increase
confidence that DFO-LS has found the global minimum, we also initiate from multiple starting points.

## 2.5   Misfit functions

Every evaluation of our misfit requires running the biogeochemical model for 3000 years before calculating the misfit between
the model outputs and synthetic observations. While both CMA-ES and DFO-LS minimise a single misfit function, DFO-LS
can exploit the structure of the misfit function. Thus, if the misfit is defined as per Equation 1, we only provide CMA-ES with
$f(\mathbf{x})$ whereas the individual $r_i(\mathbf{x})$ are supplied to DFO-LS. Here we define the $r_i$ to take into account the spatial structure
of the misfit by partitioning the ocean into previously established biome regions of similar ocean biogeochemical properties
(Henson et al., 2010; Weber et al., 2016), provided by Raffaele Bernardello (Barcelona Supercomputing Centre) via personal
communication, several of which were further split by depth at 1000 m (see Fig. 2) for a total of 19 regions. For every region
$j$, we further calculate a misfit for each of the 3 tracers $q$ (phosphate, nitrate, oxygen) used in the optimisation. The objective
$f(\mathbf{x})$ is thus composed of $19 \times 3 = 57$ terms of the form:

$$r_{qj}^{\epsilon}(\mathbf{x}) = \sqrt{\frac{V_j}{V_T}} \frac{\sqrt{\sum_{i=1}^{N}(m_{qi}(\mathbf{x}) - (o_{qi} + \epsilon_{qi}))^2 \frac{V_i^{i \in j}}{V_j}}}{\sum_{i=1}^{N} o_{qi} \frac{V_i^{i \in j}}{V_j}}, \tag{2}$$

where $m_{qi}(\mathbf{x})$ is the model solution with parameters $\mathbf{x}$ at grid point $i$ for tracer $q$ and $o_{qi}$ the corresponding observation (the
synthetic observations provided by a reference run of MOPS). Here $N$ is 52749, the total number of grid boxes in the model.
The misfit is normalised by the volume-weighted mean tracer concentration for that region and weighted, first, by individual
grid point volumes $V_i$ relative to the volume $V_j$ of region $j$ and, second, by the region's total volume relative to the global
ocean volume $V_T$. Real oceanic observations have a degree of uncertainty associated with them due to spatio-temporal of the
ocean, e.g., from small scale processes such as eddies. To account for this we add a noise term $\epsilon_{iq}$, which is the added noise





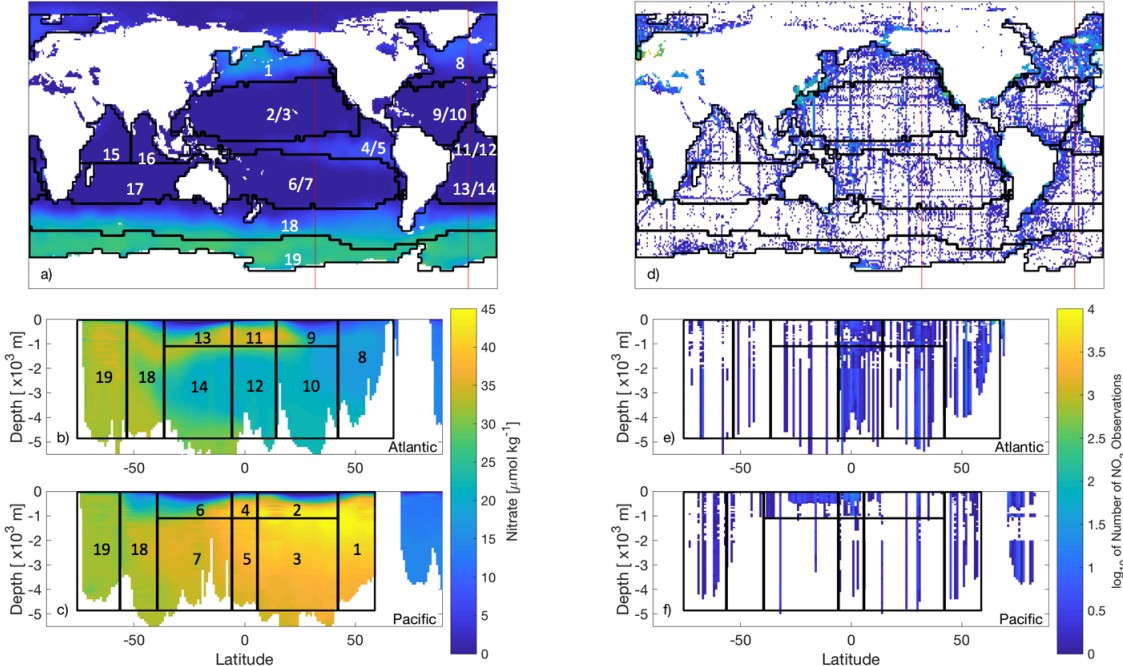

**Figure 2.** World Ocean Atlas nitrate data (Garcia et al., 2018a, b) of (left) interpolated objectively analysed mean concentration of nitrate in sea water [$\mu$mol kg$^{-1}$], and (right) number of true observations plotted on a log$_{10}$ colour scale (white oceanic areas show areas of no nitrate observations). These have been plotted for the (a,d) global surface 0 m, which also show the locations of the longitudinal transects (red lines) for the nitrate data plotted at (b,e) 23°W and (c,f) 140°W. Overlain are the boundaries of 19 biomes (Henson et al., 2010; Weber et al., 2016) of similar biogeochemistry. Regions with 2 numbers have been further split by depth.

due to uncertainty associated with tracer $q$ for every grid box $i$ in the model. The global misfit $f^{\epsilon}_{global}(\mathbf{x})$ is then defined as

$$f^{\epsilon}_{global}(\mathbf{x}) = \sum_{q=1}^{3} \sum_{j=1}^{19} r^{\epsilon}_{qj}(\mathbf{x})^2. \tag{3}$$

The total misfit function $f$ is broadly similar to Kriest et al. (2017), with the main difference being the incorporation of the 19 biome regions.

The non-noisy equivalent of these misfit terms are $r_{qj}$ and $f_{global}$ as in Equations 2 and 3 with $\epsilon = 0$. We also define "baseline" misfits $r^{Base}_{qj}$ and $f^{Base}_{global}$, which are the misfits due to the noise alone in the special case where the model outputs

equal the observations. $f^{Base}_{global}$ give an indication of the termination criteria when optimising the model to real noisy oceanic observations, as optimising below this threshold would serve no useful purpose.

To specify a realistic noise field we take the standard deviation variable provided in the World Ocean Atlas database (WOA18 Garcia et al., 2018a, b). Since our misfit is defined with respect to annual mean data we require an annual mean standard deviation without the variability of the seasonal cycle. To do so we take the numerical mean (weighted by number of observations)

of the monthly standard deviations reported in the WOA18 dataset for the upper 800 m (phosphate and nitrate) or 1500 m (oxy-





gen), and the annual standard deviation below those depths. These standard deviations fields were interpolated onto the model grid and then multiplied by three different Gaussian noise fields to create three separate noise ($\epsilon$) realisations. The baseline misfit terms $r_{qj}^{Base}$ and $f_{global}^{Base}$ were calculated as an average over these realisations.

As mentioned above, the "observations" in this study are from a reference simulation of MOPS, hereafter referred to as MOPS-ref, run with the following parameter values: $R_{O_2:P} = 170$ mmol $O_2$ : mmol P, $I_C = 24$ W m$^{-2}$, $K_{PHY} = 0.03125$ mmol P m$^{-3}$, $\mu_{ZOO} = 2$ d$^{-1}$, $k_{ZOO} = 3.2$ (mmol P m$^{-3}$)$^{-1}$ d$^{-1}$, and $b^* = 0.858$ (see Table 3).

### 2.6   Optimisation experimental design and solver settings

In this study we seek to: 1) compare the performance of CMA-ES and DFO-LS on noise-free observations; 2) investigate DFO-LS's performance on noisy observations; and 3) investigate the impact of sparse observations on the ability of DFO-

LS to recover the true parameters. In order to do so we carried out the following series of experiments (see Table 1 for the corresponding experiment labels):

***Noise-free experiments***

In the noise-free experiments we attempted to recover all 6 parameters. For this a single run of CMA-ES was performed (labelled exp_c). For DFO-LS two experiments were carried out (exp_d1 and exp_d2), starting from two different locations in

parameter space that were chosen to be relatively far from the target parameters. The parameter values for these and all other experiments are listed in Table 3.

   Both CMA-ES and DFO-LS are controlled by various solver settings. For CMA-ES the main ones are the number of sequential generations and the population size. As per Kriest et al. (2017) we set these to 200 and 10, respectively. The solver settings used by DFO-LS are summarised in Table 2. Of the noise-free experiments exp_d1 and exp_d2, the former had DFO-LS set-

tings regarding trust region management (**tr_radius**) which are more suitable for a noisy misfit function, while the latter for a smooth misfit function. As exp_d1 was slightly more successful, the trust region management settings were set to be more suitable for a noisy misfit function in all subsequent experiments.

***DFO-LS experiments with observational uncertainty***

To understand optimisation performance in the presence of observational uncertainty, noise was added to the reference obser-

vations (see Section 2.5). Three such optimisation runs, each with a different noise realisation, were carried out with DFO-LS (expd_rng1, expd_rng2, expd_rng3) starting from the same location in parameter space, to minimise the noisy misfit function $f_{global}^\epsilon$. The goal was to see if DFO-LS could recover all 6 of the MOPS-ref target parameter values.

***DFO-LS experiments with sparse observations***

There are large areas of the ocean which have not been sampled adequately or at all (e.g., Fig. 2). While it is possible to fill in

the gaps in the data using objective interpolation methods, this might not always work well in the presence of large gradients. In a last set of experiments we therefore compared how DFO-LS performs in the presence of data sparsity (exp_d1_sparse and exp_d2_sparse), by only using observations at model grid points for which the corresponding locations in WOA18 contain data, with its corresponding performance in the absence of data sparsity (exp_d1 and exp_d2).



| twin experiments tuning to: | | |
|---|---|---|
| **noise-free observations** | **noisy observations** | **sparse observations** |
| exp_c | expd_rng1 | exp_d1_sparse |
| exp_d1 | expd_rng2 | exp_d2_sparse |
| exp_d2 | expd_rng3 | |

**Table 1.** Names of each experiment tuning to noise-free, noisy and sparse twin observations. The characters within each experiment name, after "exp" follows the letter "c" or "d" depicting CMA-ES or DFO-LS. After this there is optionally a number to differentiate between different DFO-LS starting locations in parameter space, or text to identify the added noise scenario or whether the observations are sparse.

| DFO-LS Setting Name | Description | Group A* | exp_d2 |
|---|---|---|---|
| **maxfun** | Maximum number of true misfit function evaluations | 70 | 70 |
| **obj_fun_has_noise** | Does the misfit function have stochastic noise? | False | False |
| **rhobeg** | Normalised radius of parameter trust region at start | 0.1 | 0.1 |
| **rhoend** | Normalised radius of parameter trust region for termination or restart | 0.001 | 0.001 |
| **tr_radius.gamma_dec** | Ratio to decrease trust region radius ($\Delta_k$) in an unsuccessful iteration | 0.98 | 0.5 |
| **tr_radius.alpha1** | Ratio to decrease the lowest bound ($\rho_k$) for the trust region radius | 0.9 | 0.1 |
| **tr_radius.alpha2** | Ratio of $\rho_k$ to decrease $\Delta_k$ by when $\rho_k$ is reduced | 0.95 | 0.5 |

**Table 2.** DFO-LS parameter settings for each optimisation experiment. * Group A = exp_d1, t6d_fullgrid, t6d_sparse, t6d_rng1, t6d_rng2, t6d_rng3.

## 3 Results

Tables 3 and 4 show the starting and optimised parameter values, and parameter recovery information for all twin optimisation experiments. In the following sections we plot both the global misfit and parameter values for every function evaluation throughout each optimisation experiment. During one CMA-ES iteration we evaluate the misfit function 10 times (the population size). Therefore for CMA-ES we plot the minimum (best) and maximum misfits, and we plot the parameter values corresponding to the best misfit, and the minimum and maximum parameter values of each population.

To both reiterate how DFO-LS works and fully explain the DFO-LS figures, we briefly describe the optimisation process in terms of expected misfit reduction and parameter trajectories. First, DFO-LS evaluates the misfit function $n+1$ times near to the chosen starting point, therefore in the first 7 evaluations we do not expect a misfit reduction. After these initial evaluations, DFO-LS attempts to minimise the misfit function and there will be both successful evaluations (the resulting misfit is lower than previously found in the optimisation), and unsuccessful ones (the misfit is not lower). There may also be restarts, directly after

which unsuccessful evaluations are common as DFO-LS perturbs the parameter values to get out of a possible local minimum.



| Parameters | | RO2:P | IC | KPHY | muZOO | kZOO | b* | Misfit |
|---|---|---|---|---|---|---|---|---|
| Upper Bound | | 200 | 48 | 0.5 | 4 | 10 | 1.8 | |
| Lower Bound | | 150 | 4 | 0.0001 | 0.1 | 0 | 0.4 | |
| Target | | 170 | 24 | 0.03125 | 2 | 3.2 | 0.858 | 0 |
| Experiments | | | | | | | | |
| **exp_c** | Start | NA | NA | NA | NA | NA | NA | $4.231 \times 10^{-2}$ |
| | **Optimised** | **170.003** | **24.001** | **0.031** | **2.000** | **3.200** | **0.858** | **$2.909 \times 10^{-10}$** |
| **exp_d1** | | 180.000 | 40.000 | 0.100 | 2.500 | 5.000 | 0.540 | $5.248 \times 10^{-2}$ |
| | | **170.401** | **24.026** | **0.051** | **2.062** | **3.448** | **0.860** | **$4.143 \times 10^{-6}$** |
| **exp_d2** | | 190.000 | 30.000 | 0.200 | 3.500 | 1.000 | 1.100 | $2.715 \times 10^{-1}$ |
| | | **169.875** | **23.663** | **0.153** | **2.013** | **3.211** | **0.859** | **$6.747 \times 10^{-6}$** |
| **expd_rng1** | | 180.000 | 40.000 | 0.100 | 2.500 | 5.000 | 0.540 | $5.316 \times 10^{-2}$ |
| | | **170.812** | **23.856** | **0.024** | **2.531** | **5.629** | **0.870** | **$8.050 \times 10^{-4}$** |
| **expd_rng2** | | 180.000 | 40.000 | 0.100 | 2.500 | 5.000 | 0.540 | $5.316 \times 10^{-2}$ |
| | | **168.116** | **25.321** | **0.007** | **2.086** | **4.634** | **0.852** | **$8.717 \times 10^{-4}$** |
| **expd_rng3** | | 180.000 | 40.000 | 0.100 | 2.500 | 5.000 | 0.540 | $5.316 \times 10^{-2}$ |
| | | **169.234** | **23.011** | **0.053** | **2.714** | **6.352** | **0.878** | **$8.215 \times 10^{-4}$** |
| **exp_d1_sparse** | | 180.000 | 40.000 | 0.100 | 2.500 | 5.000 | 0.540 | $5.427 \times 10^{-2}$ |
| | | **169.816** | **24.002** | **0.204** | **1.689** | **2.232** | **0.854** | **$6.475 \times 10^{-5}$** |
| **exp_d2_sparse** | | 190.000 | 30.000 | 0.200 | 3.500 | 1.000 | 1.100 | $2.843 \times 10^{-1}$ |
| | | **170.022** | **23.610** | **0.150** | **2.077** | **3.415** | **0.861** | **$9.691 \times 10^{-6}$** |

**Table 3.** Results table of optimised parameters for all twin experiments. Upper section shows parameter bounds and MOPS-ref target parameters to be recovered. Lower section shows each experiment's results. Columns 2-7: (1st row) starting parameter values and (2nd row) optimised parameters for $R_{O_2:P}$ [mmol $O_2$ : mmol P], $I_C$ [W m$^{-2}$], $K_{PHY}$ [mmol P m$^{-3}$], $\mu_{ZOO}$ [d$^{-1}$], $k_{ZOO}$ [(mmol P m$^{-3}$)$^{-1}$ d$^{-1}$], and $b^*$. Column 8: (1st row) the starting global misfit and (2nd row) the lowest global misfit. NA = not applicable for CMA-ES.

Therefore on every DFO-LS figure we have plotted the misfit or parameter values for every evaluation (both successful and unsuccessful) as scattered points, and successful ones in a solid line.

On every figure of global misfits the expected baseline misfit ($f_{global}^{Base}$, see Section 2.5) is also plotted, below which any misfit reduction is within observational noise levels. On every parameter trajectory plot the "recovery zone" is also shown.
This indicates the range of parameter values within $\pm 5\%$ of the target value, normalised by the total range (upper bound minus lower bound) for that parameter. We consider a parameter to have been "recovered" by a certain number of evaluations, when all subsequent parameter values corresponding to successful evaluations remain within this recovery zone.





| Experiment | Number of evaluations required to recover parameter: | | | | | | Maximum Evaluations | Evaluation of lowest misfit | Evaluations to baseline misfit (7.0405e-04) |
|---|---|---|---|---|---|---|---|---|---|
| | RO2:P | IC | KPHY | muZOO | kZOO | b* | | | |
| exp_c | 420 | 370 | 1710 | 800 | 850 | 340 | 2000 | 1983 | 309 |
| exp_d1 | **16** | **25** | **12** | 41 | 42 | **13** | 70 | 45 | 20 |
| exp_d2 | 43 | **9** | - | 44 | **29** | **26** | 70 | 49 | 35 |
| expd_rng1 | **13** | **24** | **24** | - | - | **12** | 70 | 29 | - |
| expd_rng2 | **38** | **39** | 43 | **11** | - | **12** | 70 | 43 | - |
| expd_rng3 | **25** | **24** | 57 | - | - | **12** | 70 | 57 | - |
| exp_d1_subsel | **18** | 46 | - | - | - | **20** | 70 | 62 | 21 |
| exp_d2_subsel | **25** | **9** | - | **26** | **26** | **25** | 70 | 64 | 29 |

**Table 4.** Results table of number of evaluations required to recover each parameter, for all twin experiments. Columns 2-7: number of misfit function evaluations required to successfully recover that parameter (- = never recovered). All evaluations required to recover a parameter which were fewer than 40 are typed in bold font. Column 8: the maximum number of evaluations completed. Column 9: the evaluation which provided the lowest or "best" global misfit. Column 10: the number of evaluations needed for the global misfit to be reduced below noise levels (- = never reached the baseline).

## 3.1 Noise-free experiments

### *CMA-ES experiment*

Figure 3 shows that during the optimisation exp_c by CMA-ES the global misfit decreases significantly from $10^{-1}$ to $\approx 10^{-4}$ within the first 500 function evaluations. Subsequently progress slows down as the misfit is reduced by only one more order of magnitude over the next 1000 evaluations. Progress then significantly improves, with the misfit decreasing from $\approx 10^{-5}$ to $10^{-9}$ within the final 500 evaluations. Note that the spikes in the maximum global misfit near the $1650^{th}$ evaluation was due to the added penalty factor when one of the parameter values in this population had a value just outside of its allowed

range. While this experiment did not include noise, we note that CMA-ES required 309 evaluations to reach the baseline misfit, beyond which any misfit reduction would have been within observational noise levels.

Figure 4 shows how the 6 parameters were optimised towards the MOPS-ref target parameter values by CMA-ES. The targets were found relatively quickly within the initial 500 evaluations for the parameters $R_{\mathrm{O_2:P}}$, $I_\mathrm{C}$ and $b^*$, corresponding to the initial fast misfit reduction previously shown in Fig. 3. The $\mu_{\mathrm{ZOO}}$ and $k_{\mathrm{ZOO}}$ targets were found next after approximately

1000 evaluations, after which the optimiser began tuning $K_{\mathrm{PHY}}$ towards its target until it located after 1700 evaluations. If exp_c had been terminated once the observational noise level was reached after 309 evaluations, $R_{\mathrm{O_2:P}}$, $I_\mathrm{C}$ and $b^*$ would have been optimised to their MOPS-ref values relatively well, while $K_{\mathrm{PHY}}$, $\mu_{\mathrm{ZOO}}$ and $k_{\mathrm{ZOO}}$ would still be far from their target values.



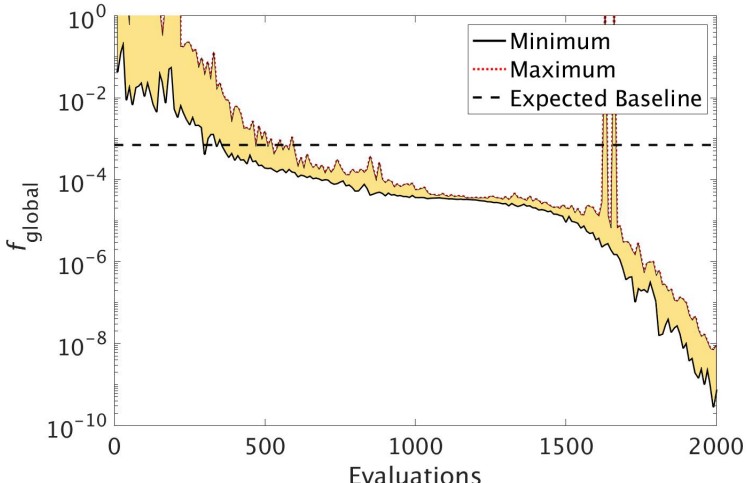

**Figure 3.** The reduction in global misfit of MOPS to the twin MOPS-ref observations by CMA-ES for the experiment exp_c. There were 10 MOPS evaluations within each CMA-ES iteration (population $\lambda = 10$), ran in parallel. Plotted is the baseline misfit (horizontal black dashed line), the minimum (black solid line) and maximum (red dotted line) misfit of each population, with the area between shaded yellow.

### DFO-LS experiments

To compare the performance of DFO-LS with CMA-ES we carried out two optimisation experiments with DFO-LS (exp_d1 and exp_d2) starting from two different locations in parameter space, with differing parameters controlling the DFO-LS trust-region shrinking speed. Optimisation exp_d1 had slower trust region shrinking parameters to allow it to better handle an irregular misfit function. Figure 5 shows the comparison between both experiments' reduction of the global misfit. In both cases there was rapid initial misfit decrease from near $10^{-1}$ to $10^{-3}$ within 30 model evaluations. Optimisation exp_d2 showed

slightly slower misfit reduction, needing 35 evaluations to reach the baseline misfit, while exp_d1 only required 20 to reach the baseline, beyond which any misfit reduction is within observational noise levels. In both exp_d1 and exp_d2 DFO-LS managed to reduce the misfit to below $10^{-5}$ within 45-49 evaluations, then initiated restarts to reduce it further. As exp_d1 performed slightly better than exp_d2, especially between evaluations 15-45, exp_d1 trust region shrinking settings were used as defaults for all subsequent experiments.

Figure 6 shows how the 6 parameters were optimised towards the MOPS-ref target parameter values by exp_d1 and exp_d2. In both experiments within the first 30 evaluations $R_{\mathrm{O_2:P}}$, $I_\mathrm{C}$, $k_\mathrm{ZOO}$ and $b^*$ were optimised to relatively close to their targets, and $\mu_\mathrm{ZOO}$ within the first 45 evaluations. $K_\mathrm{PHY}$ was successfully optimised by exp_d1, however was not successfully optimised at all by exp_d2. If exp_d1 and exp_d2 had been terminated once reaching the noise baseline, after 20 and 35 evaluations respectively, $R_{\mathrm{O_2:P}}$, $I_\mathrm{C}$, $b^*$ and $k_\mathrm{ZOO}$ would have been optimised to their MOPS-ref values relatively well.



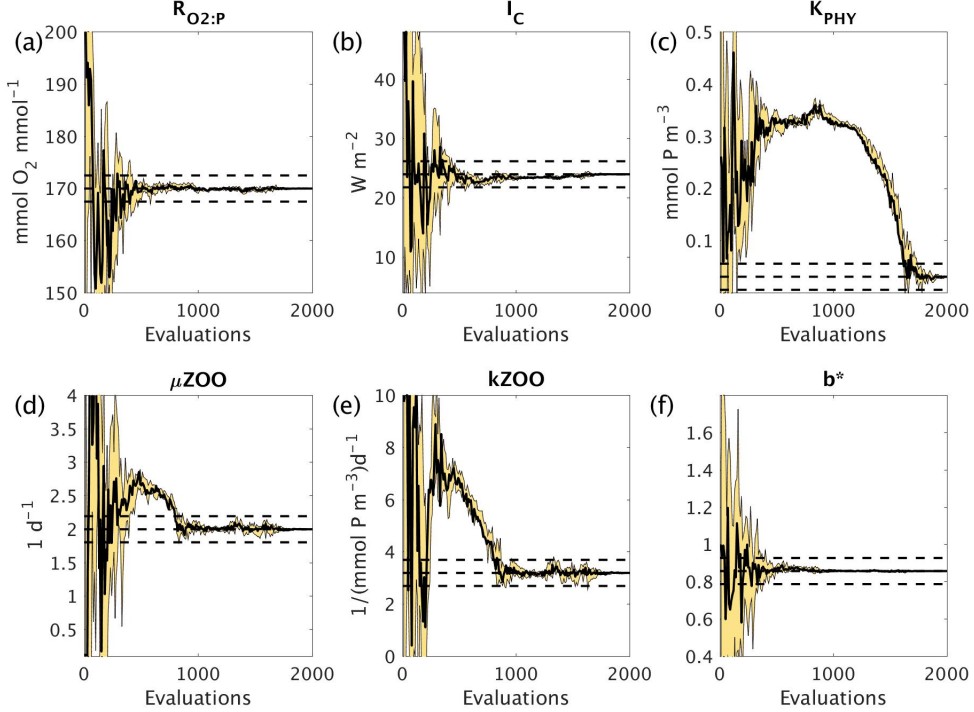

**Figure 4.** Parameter tuning by CMA-ES for the experiments exp_c for the parameters (a) $R_{O_2:P}$, (b) $I_C$, (c) $K_{PHY}$, (d) $\mu_{ZOO}$, (e) $k_{ZOO}$ and (f) $b^*$. There were 10 MOPS evaluations within each CMA-ES iteration (population $\lambda = 10$), ran in parallel. Plotted are the parameter values associated with the minimum misfit of that population (black solid line), and the maximum and minimum of all parameter values within that population (red dotted line), with the area between shaded yellow. Also plotted are the MOPS-ref target parameter values and $\pm 5\%$ recovery zone (horizontal black dashed lines).

### 3.2 DFO-LS experiments with observational uncertainty

To assess the impact of observational uncertainty we carried out three experiments in which DFO-LS was initialised from the same starting location in parameter space, but with three different realisations of random noise added to the observations (see Section 2.6). As seen in Fig. 7 DFO-LS managed to reduce the misfit to very close to the average baseline misfit within 30 evaluations with a reduction in misfit from $\sim 10^{-1}$ to $\sim 10^{-3}$. Closer to the end of the optimisation runs, restarts were initiated to encourage further misfit reduction, hence the large variations in misfit. Figure 8 shows that the initial misfit reduction corresponds to improved values for the parameters $R_{O_2:P}$, $I_C$, $K_{PHY}$ and $b^*$. DFO-LS seems to compensate for the noise by increasing the values for the parameters $\mu_{ZOO}$ and $k_{ZOO}$.

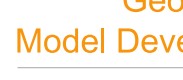
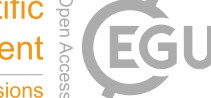

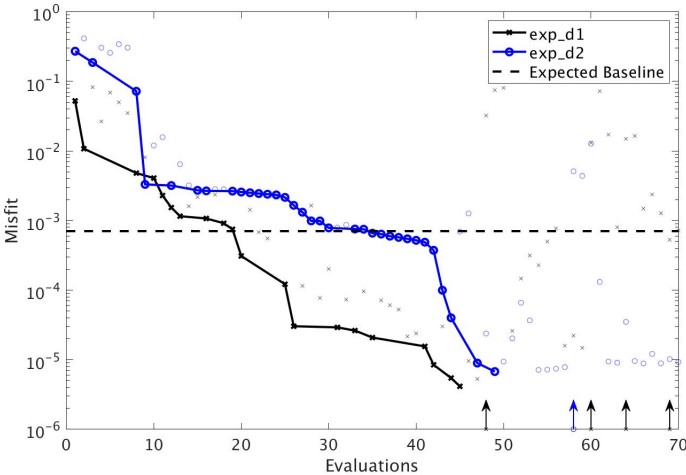

**Figure 5.** The reduction in global misfit of MOPS to the twin MOPS-ref observations by DFO-LS for the experiments exp_d1 (black line with crosses) and exp_d2 (blue line with circles). Also plotted is the baseline misfit (horizontal black dashed line). Vertical arrows indicate a soft restart, coloured and marked according to each experiment. Note that every MOPS evaluation has been plotted with a small marker, however only MOPS evaluations which resulted in a lower misfit than previously seen in each optimisation experiment has been plotted with a solid line.

## 3.3 DFO-LS experiments with sparse observations

In a final set of experiments we examine whether DFO-LS is able to successfully optimise MOPS given a sparse set of
observations (see Section 2.6). The experiments exp_d1_sparse and exp_d2_sparse were initialised from the same location in parameter space as exp_d1 and exp_d2, respectively, but the former were optimised using observations sub-sampled at grid points corresponding to locations in the un-interpolated WOA18 database. In these experiments no noise was added to the observations. Figure 9 shows that the two optimisations using full observations (exp_d1 and exp_d2) converged to slightly lower misfits than when using sparse observations. Figure 10 shows that exp_d1 successfully recovered all 6 parameters within
42 evaluations, while exp_d1_sparse only successfully recovered $R_{\mathrm{O_2:P}}$, $I_\mathrm{C}$ and $b^*$ throughout the optimisation. The above results would indicate a poorer optimisation when using sparse observations, however, from the other starting point exp_d2 successfully recovered 5 parameters within 44 evaluations, while exp_d2_sparse recovered the same 5 after only 26 evaluations. This suggests that even with sparse observations it is possible to successfully optimise a global ocean biogeochemical model such as MOPS.

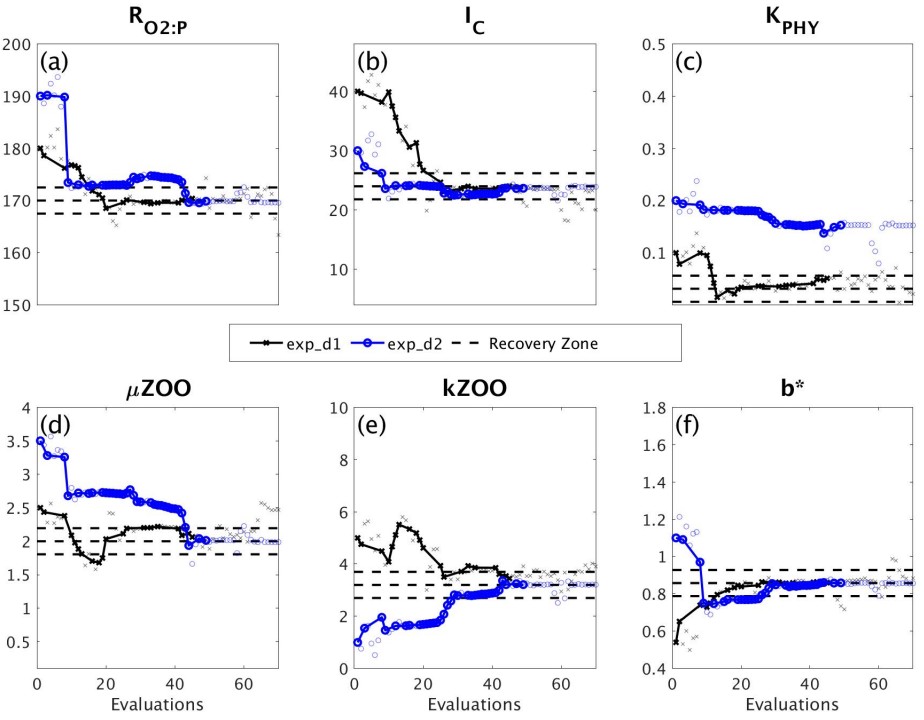

**Figure 6.** Parameter tuning by DFO-LS for the experiments exp_d1 (black line with crosses) and exp_d2 (blue line with circles) for the parameters (a) $R_{O_2:P}$, (b) $I_C$, (c) $K_{PHY}$, (d) $\mu_{ZOO}$, (e) $k_{ZOO}$ and (f) $b^*$. Also plotted are the MOPS-ref target parameter values and $\pm5\%$ recovery zone (horizontal black dashed lines). Note that every MOPS evaluation has been plotted with a small marker, however only MOPS evaluations which resulted in a lower misfit than previously seen in each optimisation experiment has been plotted with a solid line.

## 4 Discussion

### 4.1 CMA-ES vs DFO-LS optimisation performance

Our comparison of the two optimisation algorithms shows that DFO-LS could recover all 6 target parameter values within ∼40 evaluations of MOPS, while CMA-ES achieved the same goal within ∼1700 evaluations. By "recover" we mean optimised to within ±5% (normalised by the parameter range) of the target value. DFO-LS reduced the misfit to below the observational uncertainty threshold within 20-35 evaluations, while CMA-ES required 309 evaluations. DFO-LS is thus significantly more efficient for this particular problem and may, in general, be more practical for optimising more than a small handful of parameters. However we note that the multiple evaluations CMA-ES requires can be run in parallel. In contrast, DFO-LS, except for the initial $n+1$ evaluations, runs sequentially.

CMA-ES is a single-objective optimiser, while DFO-LS can use information from multiple misfit values instead of just one. Therefore it can exploit more information to allow for a faster reduction in the misfit. Neither algorithm can completely



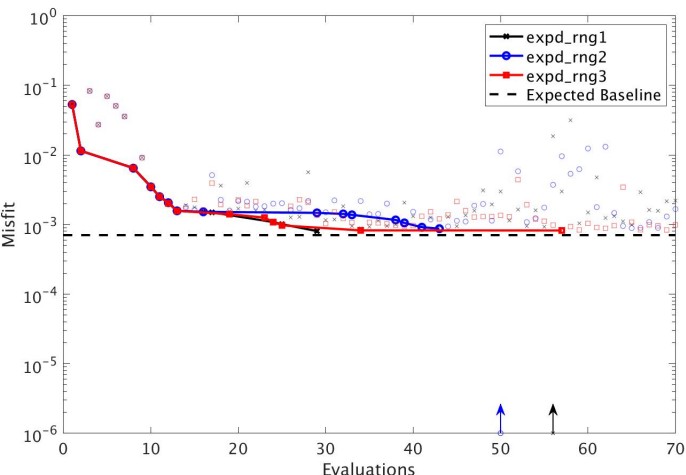

**Figure 7.** As in Fig. 5, but for experiments expd_rng1 (black line with crosses), expd_rng2 (blue line with circles), and expd_rng3 (red line with squares).

guarantee a global optimum solution, although CMA-ES carries out a more global search than DFO-LS. There is significant evidence DFO-LS can find the global optimum (Cartis et al., 2018), but to increase confidence in the final solution it can be combined with a globalising method such as starting from different points in parameter space or using the DFO-LS restart functionality.

Both methods struggled with one of the parameters $K_{\mathrm{PHY}}$, due to the misfit function's low sensitivity to this parameter (as found by perturbing the parameter values in each direction and computing the gradient). DFO-LS had not begun to tune this parameter for one of the experiments (exp_d2) before we terminated it at a maximum of 70 evaluations, although it did find $K_{\mathrm{PHY}}$ when initiated from a different starting point (exp_d1). CMA-ES also had difficulty in tuning $K_{\mathrm{PHY}}$ and only started optimising this parameter after all the other parameters were recovered at ∼1200 evaluations. The maximum number of DFO-

LS evaluations was set to 70 as it is a sequential algorithm, therefore it was impractical to allow too many more evaluations. Had it been allowed to run longer the expectation is it would begin tuning $K_{\mathrm{PHY}}$ once the other 5 were sufficiently tuned, as was the case with CMA-ES.

### 4.2    Calibrating to uncertain observations

Real oceanic observations come with associated uncertainty due to measurement error, temporal variations such as seasonal

and diurnal cycles, and meso-scale variability due to factors such as eddies and the movement of fronts. Here we have studied how this uncertainty raises the base of the misfit function, below which any optimisation of the biogeochemical model would be within the uncertainty level. We determined this baseline for the misfit function (or termination threshold) using the standard deviations of the observational data, however others have defined it as the global optimum of a surrogate formulation of the



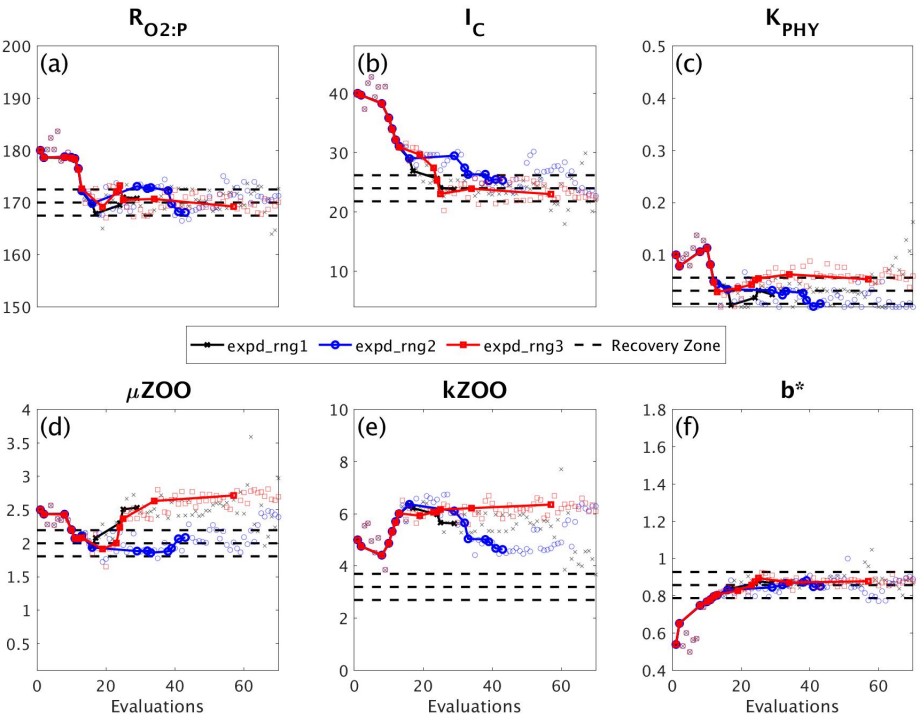

**Figure 8.** As in Fig. 6, but for the experiments expd_rng1 (black line with crosses), expd_rng2 (blue line with circles), and expd_rng3 (red line with squares).

biogeochemical model (Sauerland et al., 2017). In the present case and the chosen set of oceanic observations, the model was

significantly optimised before reaching levels of observational uncertainty, particularly due to optimisation of the parameters which the model is most sensitive to, as was determined by perturbing each parameter while holding the others fixed and calculating the misfit. In this case it was $I_{\mathrm{C}}$ (the phytoplankton half saturation for light) and $b^*$ (the increase in particle sinking speed with depth). Somewhat surprisingly, a parameter the model is less sensitive to, $R_{\mathrm{O_2:P}}$ (the ratio of oxygen consumption to phosphate release during remineralisation) was also well optimised before reaching the baseline. Despite the low sensitivity,

possibly caused by narrow parameter bounds, the high optimisation potential by this parameter may be due to the fact that the misfit function includes both oxygen and phosphate. It could also be due to the fact that this parameter has a non-local effect, as it influences the flux of oxygen and phosphate to the deeper ocean, hence to ocean basins further along the "conveyor belt". This is also the case for $b^*$ (Kwon and Primeau, 2006) but even more so as it also influences the vertical flux of all three of the tracers. In order to optimise less sensitive parameters before reaching noise levels one could introduce more metrics into the

misfit calculation to help constrain these parameters, such as phytoplankton and zooplankton data or the location of oxygen minimum zones (Niemeyer et al., 2019).

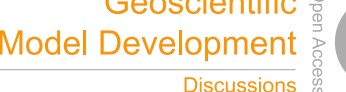

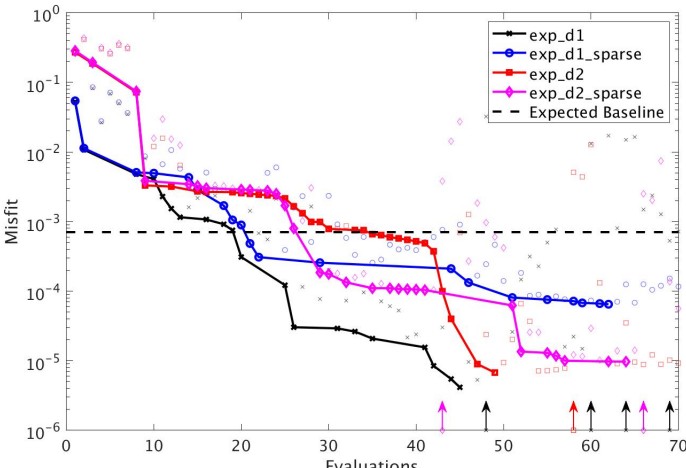

**Figure 9.** As in Fig. 5, but for experiments exp_d1 (black line with crosses), exp_d1_sparse (blue line with circles), exp_d2 (red line with squares) and exp_d2_sparse (magenta line with diamonds). Note that the baseline misfit (horizontal black dashed line) was calculated using the full grid noisy observations.

### 4.3 Calibrating to sparse observations

We also investigated the ability of DFO-LS to optimise MOPS in the presence of sparsity in the observational data. The results shown here suggest that there is no significant difference in the performance of DFO-LS when tuning to data at every grid
point verses a subset of grid points. Interpolating can introduce large errors, on the order of 20% (Garcia et al., 2018a, b), particularly in poorly sampled regions such as the Southern Ocean. However, our experiments suggest that it is possible to use un-interpolated observations, but it is important to start the optimiser from multiple locations in parameter space, or to generously allow restarts, in the presence of a complex misfit function with many local minima. These multiple runs clearly can be run in parallel.

## 5 Summary

This study compared the efficiency and performance of two derivative-free optimisation algorithms, CMA-ES and DFO-LS, applied to MOPS, a global ocean biogeochemical model with 7 prognostic tracers. The two methods were used to tune 6 of the key parameters that control the behaviour of MOPS. We found that DFO-LS has a significantly lower computational cost when compared to CMA-ES, between one and two orders of magnitude, which is important considering that global
ocean biogeochemical models are computationally expensive, as they must be integrated for several thousand years to reach equilibrium. DFO-LS exploits more information when minimising the misfit function, therefore has more scope for reducing



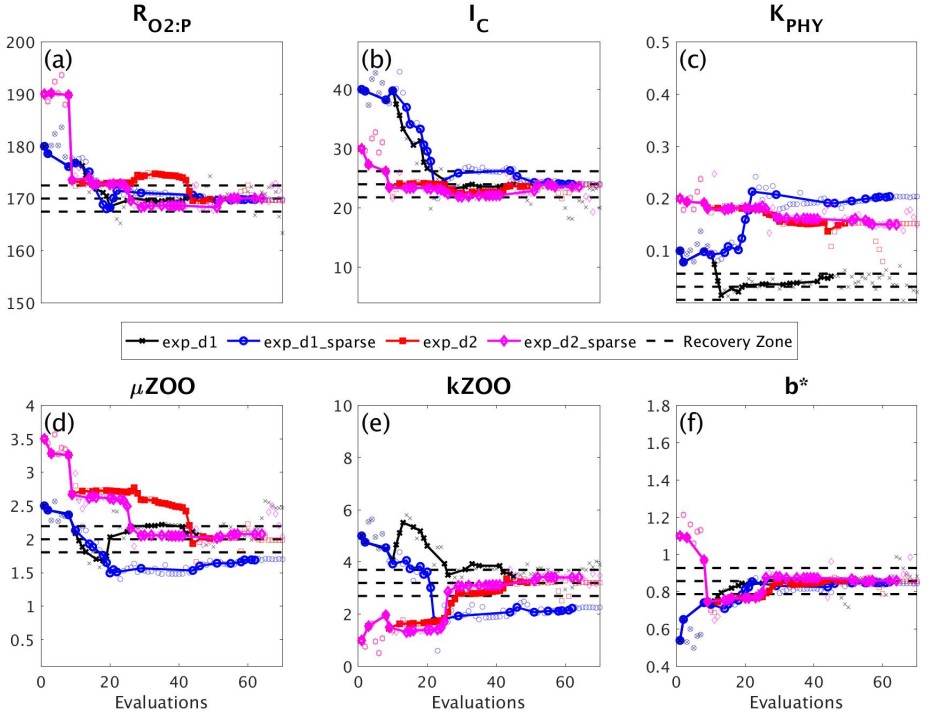

**Figure 10.** As in Fig. 6, but for the experiments exp_d1 (black line with crosses), exp_d1_sparse (blue line with circles), exp_d2 (red line with squares) and exp_d2_sparse (magenta line with diamonds).

the misfit faster than CMA-ES. However, as DFO-LS is more of a local optimiser than CMA-ES, it should be paired with a globalising method such as starting from different initial points in parameter space, which can easily be run in parallel.

Future work will involve applying DFO-LS to tune the MEDUSA biogeochemical model (Yool et al., 2011, 2013) to real
observations. MEDUSA is more typical of the biogeochemical models which are embedded within Earth System Models (in the case of MEDUSA, UKESM) that are used to project climate change.

## 6  Code Availability

The base TMM and MOPS code used for the ocean biogeochemical simulations are available to download from https://doi.org/
10.5281/zenodo.1246300. Transport matrices and forcing fields required to perform the simulations can be downloaded from
https://doi.org/10.5281/zenodo.5517238. Modifications to the MOPS code for the specific experiments described in this paper, along with model output and scripts to recreate the figures shown here, are available from https://doi.org/10.5281/zenodo.5517626. The OptClim optimisation framework used in this study to couple any climate model to any optimiser is available at





https://doi.org/10.5281/zenodo.5517610. This includes the CMA-ES optimisation code taken from the Supplement of Kriest et al. (2017) and adapted to work with OptClim.

*Acknowledgements.* Computing resources were provided by the University of Oxford Advanced Research Computing (ARC) facility (http://dx.doi.org/10.5281/zenodo.22558). SEO is grateful to the National Environmental Research Council (NE/L002612/1), the Oxford Doctoral Training Partnership in Environmental Research, and the Met Office, for studentship and funding. SK was supported by UK NERC grants NE/M020835/1 and NE/P019218/1.

*Author contributions.* SK and CC conceived the project, and together with SEO designed the experiments. SEO performed carried out the
experiments and analysis, and wrote the manuscript with contributions from all co-authors.

*Competing interests.* The authors declare that they have no conflict of interest.





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

## Appendix A: CMA-ES algorithm description

Below is a simplified description of the $(\mu/\mu_W,\lambda)$-CMA-ES algorithm (Hansen, 2016).

---

**Algorithm 1** $(\mu/\mu_W,\lambda)$-CMA-ES

---

0: INPUT: Set initial parameters as in Table 1 of Kriest et al. (2017), population size $\lambda = 10$, $\mu = \lambda/2$, evolution paths, covariance matrix C=I, distribution mean, step size and maximum generation number.

1: **while** maximum generation is not reached and fitness distribution is not flat **do**

2:     **Sample population of new probability distribution**

3:     **for** $k = 0, 1, 2, ...\lambda$ **do**

4:         Sample search point for this $k$

5:     **end for**

6:     **Update probability distribution**

7:     Update the mean of the search distribution according to a weighted average of the best half of the previously sampled population

8:     Update the overall standard deviation ("step size")

9:     Update evolution paths

10:     Update covariance matrix

11: **end while**

---





**Appendix B:  DFO-LS algorithm description**

Below is a simplified description of the DFO-LS algorithm in the context of how it has been used in this study. Not all technical

details are included, such as safeguarding steps to improve the geometry of points and the quality of the model, therefore see

the full description in Cartis et al. (2019).

---

**Algorithm 2** DFO-LS

---

**Require:** Number of parameters $n$, starting point $\mathbf{x}_0 \in \mathbb{R}^n$, minimum trust region radius ($p_{end}$), if hard or soft restarts are allowed (see
   Section 2.4), and maximum number of true misfit function evaluations.

1: Evaluate the true misfit function at $n + 1$ points within the initial trust region to build the initial interpolation set $\{\mathbf{Y}_0\}$ (this can be done
   in parallel).

2: **for** $k = 0, 1, 2, ...$ **do**

3:     **if** we have exceeded the maximum number of true misfit function evaluations **then**

4:         terminate.

5:     **end if**

6:     Construct a quadratic approximation of the true misfit function.

7:     Approximately solve the trust region subproblem to locate the minimum of the approximation within the trust region and get step $\mathbf{s}_k$
   to this point.

8:     Evaluate the true misfit function at $\mathbf{x}_k + \mathbf{s}_k$.

9:     **if** the misfit is significantly decreased **then**

10:         Accept Step:

11:         Set $\mathbf{x}_{k+1} = \mathbf{x}_k + \mathbf{s}_k$.

12:         **if** misfit decrease is not significant **then**

13:             call a hard or soft restart if allowed, or terminate.

14:         **end if**

15:         Form $\{\mathbf{Y}_{k+1}\}$ by replacing the worst point with the new accepted point to maintain a set of $n + 1$ points.

16:     **else**

17:         Reject Step:

18:         Set $\mathbf{x}_{k+1} = \mathbf{x}_k$ and shrink the trust region.

19:         **if** the trust region radius is smaller than $p_{end}$ **then**

20:             call a hard or soft restart if allowed, or terminate.

21:         **end if**

22:         Make $\{\mathbf{Y}_{k+1}\} = \{\mathbf{Y}_k\}$.

23:     **end if**

24: **end for**

---