# Peer review of "A derivative-free optimisation method for global ocean biogeochemical models"

_Geoscientific Model Development, 2021_

## Author Response (AR1)

Hello,

We thank both reviewers very much for their comments and corrections. We have responded to each reviewer individually below.

Reviewer 1

We thank the reviewer very much for their comments and corrections.

Regarding the 5 technical corrections, we thank the reviewer for these, we agree with them all and have edited accordingly.

1. "The CMA-ES subsubsection and the DFO-LS subsection have different depth levels (2.3.1 and 2.4). Using 2.3.1 and 2.3.2 (or 2.4 and 2.5) would be more consistent."

   We thank the reviewer, and have changed them to 2.4.1 and 2.4.2.

2. "The sentence beginning in line 157 seems incomplete. Is there a word missing?"

   We thank the reviewer, and we agree, there were some words missing. The sentence now reads "Real oceanic observations have a degree of uncertainty associated with them due to spatio-temporal oceanic processes, e.g., from small scale processes such as unresolved eddies."

3. "I would remove the first three words "Results table of" from the captions of Tables 3 and 4."

   We agree and have removed "Results table of".

4. "In Figure 4 the parameter boundary lines are not "red dotted" as stated in the caption but black thin lines."

   We thank the reviewer for highlighting this error, and have corrected this to "thin black lines".

5. "Line 320: "verses" => "versus""

   Thank you, we have done this.

Regarding the two questions, we have replied to each below.

1. "Did you initially work without a partition into 27 biomes and 3 tracers?"

   That is an interesting question. Essentially, because of how DFO-LS works, no we didn't start off without partitioning the misfit into regions and tracers. DFO-LS requires an input vector of misfit terms. When DFO-LS uses these terms to create the quadratic approximation, it solves a system of linear equations, which would become underdetermined if the misfit vector length were less than the number of parameters being optimised. This is not a problem for DFOLS - which then uses a Tickhonov-like regularization to the ensuing inverse

problem. Still, as we are trying to perform parameter tuning, it is better, if possible, to provide DFO-LS with at least n+1 misfit terms (n=number of parameters) as this gives the code more problem information to exploit. For the upper limit, there is not really a maximum preferred length for the misfit vector. For example, one could provide a misfit for each grid point of the ocean biogeochemical model being optimised, providing a misfit vector length of many thousands. However, the problem with this is many of the misfits physically close to each other in the model will respond very similarly to perturbations in the biogeochemical parameters being optimised, which will essentially result in a heavier weighting to this location of the ocean model. Therefore, the misfit terms should be provided in such a way that they respond to parameter perturbations independently of each other, such as by providing misfits for different observational types (e.g. nitrate, oxygen, phosphate, silicate, etc), for different spatial regions (e.g. North Atlantic, Southern Ocean, etc) and for different depths in the water column (e.g. 0-1000m, 2000-4000m, etc). This is why we initially chose 19 regions, and 3 tracers, which we didn't deviate from.

We have added the following text into Section 2.5 for clarity:

"There is no maximum suggested length of the vector r, therefore the misfit at every grid point within the model domain could be provided to DFO-LS. However, many of the individual $r_i$ misfits would be physically close to each other in the model and therefore will respond similarly to perturbations in the biogeochemical parameters being optimised, which will result in a heavier weighting to this location of the ocean model. To avoid this, we define the $r_i$ to take into account the spatial structure of the misfit by partitioning the ocean into previously established biome regions…"

2. "Does the required number of function evaluations (e.g. to reach "baseline optimality") significantly increase if the objective function is provided as a single sum of squared differences, only?"

Indeed we have tried this. We cannot use DFOLS for such an experiment, but a related code called BOBYQA (derivative free, builds a quadratic approximation, minimises within a trust region etc). BOBYQA is meant for general function minimisation, and so when one applies it to our problem, one indeed would supply only (calls to) the scalar sum of squared differences (not the individual misfit terms). We gave DFO-LS the 19x3 misfits, while we gave BOBYQA the summed square of the 19x3 misfits, and let them both run for about 70 iterations. Their success was very similar, though the parameter recovery by BOBYQA was slower than for DFO-LS by about 10 iterations (which is why we chose to continue with DFO-LS). The main paper for DFOLS [1] and references therein compared DFOLS to BOBYQA on standard data fitting test problems and found DFOLS to be significantly more efficient. Thus in our experience, there are some computational gains in supplying the misfit terms individually, if a code such as DFOLS is able to exploit them.

Reviewer 2

We thank the reviewer very much for their comments and corrections. We have made revisions as suggested in all comments made by the reviewer, with the one exception of item 47 below, where we hope the new information provided justifies this decision.

In the itemised list below we have responded to each of the suggestions and comments provided by the reviewer (in italics). We thank the reviewer for these, and hope we have responded satisfactorily and clearly.

1. *l. 1: "performance" of biogeochemical models could be confused with computational cost (the way "performance" is used when talking about DFO-LS). What about "skill"?*

   We thank the reviewer for this comment. We have changed "performance" to "skill" when discussing biogeochemical models as suggested, in Line 1 and also in Line 22.

2. *l. 34: I am unsure the Li and Primeau (2008) citation is an application of the Transport Matrix Method. (Maybe clarify or remove it?)*

   We thank the reviewer for this comment. Li and Primeau (2008) use a version of transport matrix methodology, however not the Transport Matrix Method specifically mentioned in this sentence, so rephrasing was necessary for clarity. We have changed

   "However, with the aid of fast "offline" circulation schemes, such as the Transport Matrix Method (Khatiwala et al., 2005; Li and Primeau, 2008) which can be applied to time-dependent biogeochemical models, more recently, complex global ocean biogeochemical models have also begun to be systematically optimised to observations ..."

   to be

   "However, with the aid of fast "offline" circulation schemes, such as those using transport matrix methodology (e.g. Khatiwala et al., 2005; Li and Primeau, 2008) which can be applied to time-dependent biogeochemical models, more recently, complex global ocean biogeochemical models have also begun to be systematically optimised to observations ...".

3. *l. 37: "use finite differences or adjoint" This is an inexact distinction of cases. Derivatives can sometimes be derived symbolically (manually or computationally) and evaluated directly (see, e.g., Dickinson and Gelinas, 1976; doi:10.1016/0021-9991(76)90007-3). Most often symbolic derivations are impractical, so one falls back on numerical techniques, such as finite differences. But there are other more efficient and accurate methods (see, e.g., Griewank and Walther, 2008; doi:10.1137/1.9780898717761).*

   We thank the reviewer for this comment. We have rephrased

   "Derivative-based algorithms such as Gauss-Newton (Hartley, 1961) use finite differences or adjoints to calculate derivatives within the parameter space to locate minima. They can be both computationally expensive and generally less robust on or even unsuitable for noisy problems. Derivative-free algorithms (Conn et al., 2009) in contrast can be less computationally expensive and are better adapted to handle noisy misfit functions."

   to

   "Derivative-based algorithms such as Gauss-Newton (Hartley, 1961) use derivatives within the parameter space to locate minima. The calculation of derivatives, which can be undertaken using finite differences or automatic differentiation/adjoints (Griewank and Walther, 2008), can be prohibitively expensive in some cases, such as when evaluating the misfit function is computationally costly or noisy (Chapers 8 and 9 of Nocedal and Wright, 2006). By contrast, derivative-free algorithms (Conn et al., 2009) may require less evaluations per iteration and are typically better adapted to handle noisy misfit functions."

4. *l. 37–38: "to calculate derivatives" This is technically incorrect. Gauss-Newton and other derivative-based algorithms use derivatives but do not calculate them.*

We thank the reviewer for this comment. The revision made in regard to the previous comment (item 3) satisfies this comment also.

5. *l. 38–39: "They can be both computationally expensive and generally less robust on or even unsuitable for noisy problems" Is there a reference for this? While I am convinced that the authors are correct with regards to the pitfalls of a derivative-based algorithm in the case of noisy problems, I am failing to see a strong argument for computational efficiency. While it is true that evaluating a derivative is not free, it provides information that can drastically improve the convergence rate of the optimization algorithm. This needs to be discussed in more detail in my opinion.*

We thank the reviewer for this comment. We hope the revision made in regard to the previous comment (item 3) satisfies this comment also.

We believe there's a strong argument to be made based on computational efficiency. To calculate the gradient in n-dimensional space with finite differences requires n+1 function evaluations, i.e., running the model n+1 times to equilibrium. If using automatic-differentiation (AD), the code not only requires formulating in a way that is differentiable (which would entail rewriting many widely used, existing models), but for the adjoint of the full ocean circulation-biogeochemical model to be constructed, along with the associated complexities of check pointing (saving) the full nonlinear model trajectory over a 3000-year spinup. None of this is trivial or cheap, and is impractical to implement for most existing ocean/biogeochemical models. Lastly, we note that AD requires appropriate software. Nearly all biogeochemical models are written in Fortran, and in our experience the only reliable AD tool in this topic area is TAF, a commercial product. We could add a footnote to include this, if the reviewer feels this is necessary.

6. *l. 58: "interpolated" I think one could argue that the authors here mean extrapolated.*

We thank the reviewer for this comment. When mapping global sparse observations to a global grid, it can be done mainly by interpolation, and with some extrapolation (maybe at the coastlines and near the sea floor). We have rephrased

"Sparse scattered oceanic observations are commonly interpolated onto a regular grid, introducing significant error, especially in regions such as the Southern Ocean with poor data coverage."

to

"Sparse scattered oceanic observations are commonly mapped onto a regular grid using methods such as objective interpolation, introducing significant error, especially in regions such as the Southern Ocean with poor data coverage."

7. *l. 60 "Section 4" and "Section 5" should be spelled out for consistency.*

We thank the reviewer for this comment. We have changed to "Section 4" and "Section 5".

8. *l. 73: Are the 6 parameters the same as those optimized by Kriest et al. (2017)? If so, this could be made clearer.*

   We thank the reviewer for this comment. We have rephrased

   "The behaviour of MOPS is controlled by several parameters, of which we have chosen 6 to consider for calibration, based on the previous optimisation study by Kriest et al. (2017). The detailed definitions and possible ranges of these parameters are described in that paper."
   to

   "The behaviour of MOPS is controlled by several parameters, of which we have chosen the same 6 parameters to consider for calibration as chosen in the previous optimisation study by Kriest et al. (2017). The detailed definitions and possible ranges of these parameters are described in that paper."

9. *l. 86: "In general" Is there a review to reference here?*

   We thank the reviewer for this comment. We have rephrased

   "In general the misfit "landscapes" of biogeochemical models tend to be quite irregular, with many local minima."

   to

   "In general the misfit ``landscapes'' of biogeochemical models tend to be nonlinear, as found by Kriest et al. (2017) for example, who converged to multiple local minima."

10. *l. 86: "quite" is unnecessary*

    We thank the reviewer for this comment. The revision made in regard to the previous comment satisfies this comment also.

11. *l. 87–88: I find the "To determine (...) synthetic observations." sentence hard to parse. Maybe it can be reworded for clarity?*

    We thank the reviewer for this comment. We have rephrased

    "To determine if an optimiser can find the global minimum within the misfit function landscape "twin" experiments are used, whereby the misfit is calculated between the model outputs and synthetic observations."

    to

    " "Twin" experiments are used to determine if an optimiser can find the global minimum within the misfit function landscape, whereby the misfit is calculated between the model outputs and synthetic observations."

12. *l. 89–90: Suggestion: "the global minimum [is zero] and optimal parameter values are known"*

We thank the reviewer for this comment. We have rephrased as suggested.

13. *l. 90: Suggestion: "We compare the performance (...)"*

We thank the reviewer for this comment. We have rephrased as suggested.

14. *l. 95: Why not use a differentiable bijection mapping the range to the real line?*

We thank the reviewer for this comment. To remain consistent with (and therefore comparable to) the previous study by Kriest et al. (2017), we applied the same methodology to encourage CMA-ES to converge within specified bounds. This happened to be a penalty score, but other methods are available, such as the one the reviewer suggests. For clarity, we have rephrased

"To ensure that parameters lie within reasonable bounds, a penalty score is added to the misfit when any parameter value goes outside of their specified range."

to

"To ensure that parameters lie within reasonable bounds, a penalty score is added to the misfit when any parameter value goes outside of their specified range, as also done by Kriest et al. (2017)."

15. *l. 96: Can "various" be replaced with a more descriptive term? Maybe "randomized"?*

We thank the reviewer for this comment, and have changed "various" to "randomized".

16. *l. 96: It might be useful to briefly describe the covariance matrix there.*

We thank the reviewer for this comment. Section 2.4.1 has been revised to satisfy this.

17. *l. 96–98: "It returns only the parameter configurations which provide the best misfits to a multivariate normal distribution of parameters, then in the next iteration it randomly draws several more parameter configurations, and repeats" is unclear. What is "it"? Which "multivariate normal distribution"? How many is "several"? This sentence sounds like the description of a brute-force random search, which paints an unfavourable picture of what CMA-ES actually does.*

We thank the reviewer for this comment. Section 2.4.1 has been revised to satisfy this. We now state:

"During each iteration, a population size of λ biogeochemical parameter vectors are sampled from a multi-variate normal distribution, which is fully described by a mean and a positive definite matrix of covariances. CMA-ES then requires the misfit function to be evaluated at these λ locations in the parameter space. The results of these are used to update the mean and covariance of the multi-variate normal distribution, before another λ biogeochemical parameter vectors are sampled for the next iteration."

18. *l. 98: What is a "population"?*

We thank the reviewer for this comment. In every iteration of CMA-ES, the misfit is

evaluated λ times (where λ is the population size of 10 in our and Kriest et al. (2017)'s studies). Section 2.4.1 has been revised to clarify this (see response to previous comment).

19. *l. 99: "and therefore aim" reverses the logic. The "aim" is to converge towards the best estimation from the onset. "Therefore", the algorithm employs the strategy to "move" towards lower misfit values.*

    We thank the reviewer for this comment. Section 2.4.1 has been revised to satisfy this. We now state:

    "With each iteration the population should be guided towards areas of the parameter landscape which provide lower expected misfit values, aiming to converge on the parameter configurations which provide the best misfits."

20. *l. 102–103: This "In order (...) in practice" sentence could be rearranged. Also, an indication of how many function evaluations would be useful. (And "quite" is unnecessary.) Suggestion: "In practice, CMA-ES requires thousands of function evaluations (...)"*

    We thank the reviewer for this comment. Section 2.4.1 has been revised to satisfy this. We now state:

    "In order to achieve this, CMA-ES can require thousands of function evaluations (e.g. 950-3460 required by Kriest et al., 2017)."

21. *l. 104: What does "was sourced" mean? Is that the exact code? Is it archived and publicly accessible?*

    We thank the reviewer for this comment. The CMA-ES code was provided by the supplementary material of the Kriest et al. (2017) study (publicly available), which we then slightly modified to make compatible with our optimisation framework. Our modified CMA-ES code is archived and publicly available (see the final Zenodo DOI in Section 6: Code Availability). Section 2.4.1 has been revised to clarify this. We now state:

    "The optimisation code was sourced from the supplementary material by Kriest et al. (2017), with some editing to make it compatible with our chosen optimisation framework (see Section 6)."

22. *l. 109: If $x$ is bounded, then starting with $x \in R^n$ is misleading. What about: "$x \in D$ a bounded domain of $R^n$"?*

    We thank the reviewer for this comment. We have revised (what was originally Line 109) to state "…where x is an n-dimensional vector of parameters, each of which is constrained within specified bounds" and in Equation 1 and equation description we now use $x \in D$ a bounded domain of $R^n$, instead of $x \in R^n$, as suggested by the reviewer.

23. *l. 109–111: Suggestion: "DFO-LS can take into account individual terms of the misfit function and use their structure to improve convergence"*

    We thank the reviewer for this comment. This has been changed as suggested.

24. *l. 114: "provably": If there is a convergence proof, then it should be cited. Unless this was supposed to be "probably"?*

We thank the reviewer for this comment. We now include a citation to Appendix A.2 in [Cartis et al 2019-DFOLS paper] that details convergence and complexity rates.

25. *l. 117: What is a typical n?*

We thank the reviewer for this comment. As *n* is the number of parameters to be optimised by DFO-LS, *n* can vary greatly depending on the number of tunable parameters within the chosen model. For clarification in the text, we now state:

"…and at *n* additional locations nearby (where *n* is the number of parameters to be optimised)…."

26. *l. 117: "for a total of n + 1 function evaluations". I think this can safely be removed.*

We thank the reviewer for this comment. This has been changed as suggested.

27. *l. 117: What does "nearby" mean?*

We thank the reviewer for this comment. During the initial sampling, the *n* points are chosen to lie "nearby" the starting location in parameter space, specifically within the initial trust region which encircles the starting location. The size of the initial trust region is determined by the DFO-LS setting **rhobeg** (see what was originally Table 2), which we have set to 0.1. This means no initial sampled point can lie further than +-10% of each parameter range from the starting point.

As we have now also moved Table 2 to the appendix and cited readers to view the DFO-LS manual for further understanding of DFO-LS settings (see comment response 43 below), we have now revised what was originally Line 117 to be:

"…and at *n* additional locations nearby (where *n* is the number of parameters to be optimised), with their proximity determined by DFO-LS settings (see Table A1)."

28. *l. 118: Suggestion: "only one misfit function evaluation is needed"*

We thank the reviewer for this comment. This has been revised alongside comment response 30 below.

29. *Fig. 1 Caption:*
    a. *How is the information "combined"? Are the squared misfits simply summed over? If that's the case, it should be stated as such. Otherwise, maybe some clarification of what goes on would be useful.*

    We thank the reviewer for this comment. The linearised misfits (green lines in Figure 1), are squared and summed, yes. We have revised the caption to state:

    "…3) These linearised misfits are then squared and summed over to give a quadratic approximation (blue line) to the true misfit function. 4)…"

b. *How can the misfit function be "evaluated if accepted". It seems this is the other way around.*

We thank the reviewer for this comment. The reviewer is correct, and this should be the other way around. We have revised the caption to state:

"…4) Within the trust-region (shaded in yellow) the minimum of the approximation is found, at which the true misfit function is evaluated. 5) If the new point is accepted, this new information is used to update the mini-local regressions, else it is rejected and the trust-region is shrunk. Steps 2-5 are then repeated …"

30. *l. 119–121: This is only important if the initial sampling constitutes the bulk of the computation. Is that the case in general?*

We thank the reviewer for this comment. The cost of initial sampling is n+1 evaluations, then each subsequent iteration usually requires only 1 evaluation (excluding geometry-improving points and restarts), and only a few of these subsequent iterations are needed to achieve some significant misfit reduction. Therefore, the initial sampling can constitute the bulk of computation, if the user only seeks significant misfit reduction. However, in our case with twin experiments, we are seeking a misfit close to zero, and therefore we continue to run more iterations and associated evaluations, meaning the initial sampling is superseded by the total number of iterations.

We are not trying to state that the cost of DFOLS increases linearly with n, only the initial sampling. We see this is only important information if the initial sampling dominates the computation expense, which we have explained above is not always the case. Therefore, we have revised:

"In subsequent iterations only one function evaluation is needed. This is important to note because it means the computational expense of the initial sampling by DFO-LS increases only linearly with the number of parameters to be optimised."

to

"In subsequent iterations typically only one function evaluation is needed, and often only a handful are needed to achieve significant misfit reduction."

31. *l. 133: "minima ." (space before dot)*

We thank the reviewer for this comment. This has been corrected.

32. *l. 139: "much" is not needed.*

We thank the reviewer for this comment. This has been corrected.

33. *l. 139: "This is much more computationally expensive than a soft restart" Why is it the case?*

We thank the reviewer for this comment. A restart is where the trust region expands, n+1 points are taken from within this new trust region, before the optimisation continues again. The difference between a hard and soft restart is how DFO-LS gets the misfit information for these new n+1 points. A hard restart is like initialising all over again, whereby the

(expensive) true misfit function is evaluated n+1 times. Contrastingly, a soft restart requires a smaller number of evaluations of the true misfit function. It "shifts" some of the existing n+1 points to get the predicted misfit information (see geometry improving points in Section 3.1 of the Cartis et al 2019-DFOLS paper).

We were not very clear with this in the text. Therefore, we have revised to state

"During a restart the trust region expands, allowing DFO-LS to search for points potentially outside the local minimum it may be trapped in, and move towards a lower minimum elsewhere. This can be done by either a "hard" restart, whereby the (expensive) misfit function is re-evaluated at n+1 new locations within the expanded trust region, or by a "sort" restart, whereby DFO-LS only "shifts" some of the current n+1 points in parameter space to geometry-improving points (Cartis et al., 2019). The former is more computationally expensive, therefore we don't use it here, although soft restarts are allowed..."

34. *l. 148–149: Are these personally communicated regions available in a public archive? Reproducibility hinges on such availability.*

We thank the reviewer for this comment. Yes, the biomes mask files are available in the third DOI given in Section 6 Code Availability.

35. *Eq. (2): Does $V_i^{i \in j}$ = 0 when $i \notin j$? If so, I would suggest just having $V_i$ instead, and summing over only $i \in j$ (instead of summing, potentially redundantly, over all i)*

We thank the reviewer for this comment. This has been revised as suggested.

36. *l. 158: "eddies" can be large. Maybe "unresolved eddies" is clearer?*

We thank the reviewer for this comment. This has been revised as suggested.

37. *Fig 2. Caption: Which reference defines the 19 regions? Henson et al (2010) or Weber et al (2016)?*

We thank the reviewer for this comment. The majority of the regions were as in Henson et al (2010), but with the regions in the Southern Ocean determined as in Weber et al (2016). To clarify this, we have revised the caption to state:

"…Overlain are the boundaries of 13 biomes of similar biogeochemistry, the majority of which were determined as in Henson et al. (2010), while those in the Southern Ocean as in Weber et al. (2016). Six regions have been further split by depth, leading to a total of 19 regions."

38. *Eq. (30) + many lines: "$f_{global}$". Usually non-variable subscripts are typeset upright. Also, "global" is misleading, since there are many tracers. On the other hand, for volumes, the authors use "T", for "total", I guess. Maybe no subscript for this "total" f? Or maybe swap the "T" and "global" subscripts throughout?*

We thank the reviewer for this comment. "T" and "global" subscripts have been swapped throughout, both in typeset upright, as suggested by the reviewer.

39. *l. 171: How is the interpolation done?*

We thank the reviewer for this comment. This was done by linear interpolation. We have revised the text to state:

"These standard deviations fields were linearly interpolated onto the model grid…"

40. *l. 172: Why three noise realizations?*

We thank the reviewer for this comment. As the noise realisations involve randomness, we had to run more than one to show robustness of results, and we could do no more than three due to computational expense.

41. *l. 189 and throughout: I kept going back to read what differentiated each experiment from the other. Maybe the authors can find more expressive names for their experiments? E.g., `exp_d1` could be `D_noise` and `exp_d2` could be `D_smooth`? `exp_drngi` could be `D_randi`? And `exp_d1_sparse` could be `D_smooth_sparse`, and so on.*

We thank the reviewer for this comment. We have revised experiment names in all tables, figures and text to the following:
- C_smooth
- D_smooth_1 and D_smooth_2
- D_noise_rand1, D_noise_rand2 and D_noise_rand3
- D_smooth_sparse_1 and D_smooth_sparse_2

42. *l. 183–204: What about an experiment combining sparse and randomized observations?*

We thank the reviewer for this comment. We found including data sparsity with non-noisy observations had little influence on convergence (excluding the insensitive KPHY, which we discuss latter in the responses to the reviewer's comments), therefore we assumed including data sparsity with the randomized observations would also have little influence. The computational expense of additional runs to confirm/deny this was deemed too costly for us to address this additional question in our study.

43. *Table 2. It is unclear what all the settings do. Also, all the caption experiment names do not match the main text.*

We thank the reviewer for these comments. The caption has been corrected. The settings in this table include too high a level of detail for the main text, therefore this table has been moved to the appendix. It is mainly for DFO-LS users, and only in case they want to know. Not knowing these settings does not hinder the reader's understanding of this study, but more in-depth descriptions can be found in the DFO-LS user manual, freely downloadable with the software. Also, they are not required to know for reproducibility, as the files in the provided DOIs already include this information, in the format required for the optimisation framework to parse.

44. *l. 208: the authors say they "plot" values but instead show a table.*

We thank the reviewer for this comment. The text first states we show Tables 3 and 4, but we do not describe these tables here in the text yet (the tables are summaries of Figures 3-10, therefore we describe them together in Sections 3.1-3.3). We then move on in the text

to state that in the following sections we then plot the global misfit and parameter values throughout each separate experiment (meaning Figures 3-10). We then explain how to understand the information in the figures, before going on to describe all the results (tables and figures together). Tables 3 and 4 are now in the appendix, and replaced by two figures (see below), therefore we have revised this text to state:

"The results for all twin optimisation experiments are summarised in Figures 4 and 5, and in Appendix C (Tables C1 and C2), which show the starting and optimised parameter values, and parameter recovery information. In the subsequent sections we then plot both the global misfit and parameter values for every function evaluation throughout each individual optimisation experiment …"

*Indeed, Table 3 looks like it would deserve to be turned into a plot with 6 panels (6x1, one for each parameter) with each experiment on the x-axis, and the optimized value on the y axis. With a color code that conveys the groupings (smooth, sparse, noisy, and so on).*

We thank the reviewer for these comments. Table 3 has been replaced by a figure such as described and suggested by the reviewer. So to not lose the precise value information of both the parameter and misfit information provided by Table 3, it has been moved to the appendix.

*The same goes for Table 4, which could be turned into a combination of bar plots (with a broken axis to cater for a large number of evaluations for CMA-ES rather than a misleading logscale).*

We thank the reviewer for this comment. Table 4 has been replaced by a figure such as described and suggested by the reviewer. Table 4 has been moved to the appendix.

45. *Table 4. Some of the experiment names do not match the main text. Is "subsel" the same as "sparse"?*

We thank the reviewer for this comment. This has been corrected.

46. *l. 277: Maybe I read this incorrectly, but it seems the authors report that only 1/6 of the DFO-LS experiments recovered all 6 target parameters (Table 4). This is swept under the rug here. I think it would be useful to discuss the failures of convergence for some parameters here.*

We thank the reviewer for this comment.

As we compare a smooth, non-sparse CMA-ES experiment to DFO-LS, we should only compare the DFO-LS experiments which were also smooth and non-sparse. These are experiments exp_d1 and exp_d2 (now called D_smooth_1 and D_smooth_2). Of these two, half recovered all 6 target parameters. Of the remaining 5 DFO-LS experiments, where noise or sparsity were included, none recover all 6 target parameters. However, these experiments were all started from the same parameter location as either D_smooth_1 or D_smooth_2. DFO-LS is deterministic, meaning the convergence from the same starting location should be the same every time, therefore any worsening in each respective experiment's convergence is solely due to the influence of either observational noise or data sparsity. Table 1 has been revised to better partition the experiments.

That being said, a half convergence success is still not ideal, and does not indicate DFO-LS can robustly recover all 6 of these parameters. However, both D_smooth_1 and D_smooth_2 experiments (the only 2 experiments we did that are comparable to the CMA-ES run) successfully recovered 5/6 of the parameters, and it was only one particular parameter which failed to be recovered: $K_{phy}$.

Our misfit function is particularly insensitive to $K_{phy}$, as we illustrate next. CMA-ES also struggled to find this parameter, as it tuned it last (after ~1700 evaluations, when all the others had been recovered by ~900 evaluations). To show this insensitivity, we have calculated the change in misfit after a 10% perturbation of each individual parameter (see figure below). A perturbation in $K_{phy}$ caused the smallest change in the misfit ($4.1 \times 10^{-6}$).

[Figure]

Therefore, we have revised to ensure our narrative states that DFO-LS can robustly recover all parameters, providing the misfit function is sufficiently sensitive to each parameter, and we have included this sensitivity information in the results section of the paper.

The alternative would be to run additional experiments with randomised starting locations, and see how often $K_{phy}$ can be successfully recovered, as the reviewer suggests in their next comment. However, the misfit function is so insensitive to this parameter, and D_smooth_2 still manages to reduce with misfit to well below the expected baseline despite unsuccessful $K_{phy}$ convergence. Therefore, we do not think the large computational expense of these additional runs is worth tuning what could be deemed a relatively unimportant parameter.

47. *l. 290–297: This seems like a significant caveat. The justification for capping the number of evaluations for DFO-LS to 70 is unsubstantiated. Combined with the above comment it seems that the authors' (5?) original experiments with DFO-LS all failed to recover all 6 parameters, and they then added an extra experiment with a different starting point for K_PHY that is at least twice as close as its target value (Figure 6). In my opinion, this places the robustness of the approach under question, and therefore I would recommend additional experiments with randomized starting values.*

We thank the reviewer for this comment. The maximum number of evaluations was determined as (n+1)*10, where n = 6 parameters (roughly equivalent to 10 iterations of a derivative based method). The 10 was subjectively chosen, as a trade-off between sufficient evaluations to allow significant misfit reduction, and computational expense of each evaluation. This value is a typical example of "short budgets" for DFO test in nonlinear optimisation. More technically, and specific to our particular case, the maximum wall clock time allowed on the supercomputing cluster used is 120 hours, within which we could just fit

70 sequential evaluations of the misfit function.

Regarding $K_{phy}$ and robustness, please see the response to the reviewer's previous comment.

48. *l. 315: While technically feasible, I would consider using "oxygen concentrations" as a constraint rather than the "location of oxygen minimum zones", which is subjectively defined by an arbitrary threshold.*
We thank the reviewer for this comment. We mean by this sentence that one could add more constraints to the misfit, such as phytoplankton and zooplankton data, or additional oxygen constrains (as we already use oxygen concentrations in our misfit) such as done by Niemeyer et al. 2019 who included the location of oxygen minimum zones. In this study they calculated the overlap between simulated and observed oxygen minimum zones. We have revised the text to state:

"In order to optimise less sensitive parameters before reaching noise levels one could introduce more metrics into the misfit calculation to help constrain these parameters, for example phytoplankton and zooplankton data, or additional oxygen constrains, such as the location of oxygen minimum zones as done by (Niemeyer et al., 2019)."

49. *Figures 7, 9 captions should repeat what the vertical arrows mean (soft restarts)*

We thank the reviewer for this comment. This has been revised as suggested.

50. *One of the main citations of the manuscript (Cartis et al., arXiv, 2018) should be replaced by the more recent and, importantly, peer-reviewed, version (Cartis et al., Optimization, 2021, doi: 10.1080/02331934.2021.1883015).*

We thank the reviewer for this comment. This has been corrected.

---

## Referee Report (RR1)

**Review (round 2) gmd-2021-175**

*Author: Benoît Pasquier*
* * *
I have found the responses and revisions of the authors appropriate and **recommend publication after some minor revisions**, listed below. (Apologies for the likely frustrating notation comments.)

- I should have been clearer in my first review point 38:

  > Eq. (30) + many lines: "$f_{global}$". Usually non-variable subscripts are typeset upright.

  This comment should have mentioned all non-variable subscript and superscript. E.g., I would replace `f^{Base}_\textrm{T}` and `r^{Base}_{qj}` with `f^\textrm{base}_\textrm{T}` and `r^\textrm{base}_{qj}` (i.e., $f^{Base}$ and $r^{Base}$ with $f^{base}$ and $r^{base}$) where I also avoided having a capital "B" for consistency. The authors should check the entire manuscript for any non-variable subscript/superscript and correct them.

- As per GMD's guidelines (https://www.geoscientific-model-development.net/submission.html#math) vectors such as **x**, which I guess was LaTeX'd from `\mathbf{x}` (first appearance l. 117) should be typeset "in boldface italics", i.e., *x*. This is easily done using the `\vec{x}` command provided by the Copernicus LaTeX template.

- Units are missing in almost every figure and should be added.

- l. 178: Eq. (2): the sums should not start at $i$=1. They should be written as `\sum_{i\in j}` instead of `\sum_{i=1}^{i\in j}` ($\sum_{i \in j}$ instead of $\sum_{i=1}^{i \in j}$).

- l. 132: Mathematical symbol D should be in italics, i.e., *D*. (Use "$D$".)

- l. 168: "the length of the vector **r**" should be spelled out for clarity with maybe something like "*d*, the number of $r_i$ terms". (Also note that otherwise the vector **r**, which should be boldface italic, is not even defined.)

- Table 1: In retrospect, my suggestions for experiment names were not great. For readability, I think it might be better to have shorter names and avoid underscores. What about:

  - "CTL" for the CMA-ES run (the control run),
  - "SMOOTH$_1$" for the "D_smooth_1 run" and so on,
  - "NOISY$_1$" for the "D_noise_rand1" run and so on, and
  - "SPARSE$_1$" for the "D_smooth_sparse_1" run and so on?

- l. 221: Use the `\times` symbol ("×") rather than the letter "x".

- Fig. 4:

  - Maybe a line for the target value could be added in the background? (and a ±5% band?)
  - Maybe show the 10 CTL (C_smooth) starting points as tiny dots?
  - There is a lot of unused vertical space in each panel. Maybe the $y$-axis limits can be tightened a bit? E.g., Fig. 4c shows maximum K_PHY values of about 0.2, but the $y$-axis goes up to 0.5.

- - The legend could be simplified to only say that circles are starting points and crosses are optimized values? (Maybe use black for the legend and then give a different color than black for the smooth values.
  - Speaking of color, a color-blind-friendly palette could be used here instead of plain black, red, and blue (e.g., colorbrewer's qualitative colors (https://colorbrewer2.org/#type=qualitative&scheme=Dark2&n=3), but there are many others!)
  - The legend could be placed at the bottom rather than in the middle to avoid visually breaking the x-axis alignments of top and bottom panels.

- Fig. 5: This is a key figure that was added in response to the 1st round of review to replace the now Table C. Yet, the main message — that DFO-LS requires much less evaluations than CMA-ES — is now obfuscated by the use of different scales and 2 *y*-axes. Better to show both on the same scale and let the visual speak for itself! The broken-axis suggestion (from the 1st review round) was not used, although it would make this much clearer in my opinion. Here is what I had in mind, e.g., for Fig. 5a (The red dashed line shows the imposed limit on evaluations for DFO-LS runs.):

[Figure]

[Figure]

I understand MATLAB is not suited for broken-axis plots, so to be helpful I have provided below the python code that produces the broken-axis plot shown above. This code can easily be used as a template to reproduce each panel in Fig. 5. Python code:

```python
import numpy as np
import matplotlib.pyplot as plt

np.random.seed(19680801)

experiment_names = ["CTL", "SMOOTH1", "SMOOTH2", "NOISY1", "NOISY2",
"NOISY3", "SPARSE1", "SPARSE2"]
nevals_to_basline_misfits = [309,20,35,70,70,70,21,29]

**If we were to simply plot pts, we'd lose most of the interesting**
**details due to the outliers. So let's 'break' or 'cut-out' the y-**
```

```
axis
**into two portions - use the top (ax1) for the outliers, and the**
bottom
**(ax2) for the details of the majority of our data**
fig, (ax1, ax2) = plt.subplots(2, 1, sharex=True)
fig.subplots_adjust(hspace=0.1)  # adjust space between axes

**plot the same data on both axes**
ax1.bar(experiment_names, nevals_to_basline_misfits)
ax2.bar(experiment_names, nevals_to_basline_misfits)

**zoom-in / limit the view to different portions of the data**
ax1.set_ylim(85, 350)  # outliers only
ax2.set_ylim(0, 85)  # most of the data

**hide the spines between ax and ax2**
ax1.spines.bottom.set_visible(False)
ax2.spines.top.set_visible(False)
ax1.xaxis.tick_top()
ax1.tick_params(labeltop=False)  # don't put tick labels at the top
ax2.xaxis.tick_bottom()

**Now, let's turn towards the cut-out slanted lines.**
**We create line objects in axes coordinates, in which (0,0), (0,1),**
**(1,0), and (1,1) are the four corners of the axes.**
**The slanted lines themselves are markers at those locations, such**
that the
**lines keep their angle and position, independent of the axes size or**
scale
**Finally, we need to disable clipping.**

d = .5  # proportion of vertical to horizontal extent of the slanted
line
kwargs = dict(marker=[(-1, -d), (1, d)], markersize=12,
              linestyle="none", color='k', mec='k', mew=1,
clip_on=False)
ax1.plot([0, 1], [0, 0], transform=ax1.transAxes, **kwargs)
ax2.plot([0, 1], [1, 1], transform=ax2.transAxes, **kwargs)

plt.axhline(y=70, linestyle=":", color="red")

plt.xticks(rotation=90, ha='center', va='top')

plt.suptitle("Number of evaluations to baseline misfit")

plt.show()
```

- Note 1: Do not pay too much attention to the code comments in the snippet above because they are from the matplotlib broken-axis example (https://matplotlib.org/stable/gallery/subplots_axes_and_figures/broken_axis.html) from which this code was slightly edited.

- Note 2: No local python installation is needded: To produce the plot above this code was edited and ran online using Binder ([https://mybinder.org/](https://mybinder.org/)).

---

## Author Response (AR2)

We thank the reviewers for their comments and suggestions. Compiled below are responses to both reviewers' comments (in italics).

Reviewer 1

*"Trying to reproduce experiment "exp_d1" (former terminology) I noticed one difference concerning the initial value for parameter "detmartin" between the provided configuration file "OxfordMOPS_exp_d1.json" (where the value is 0.5) and the provided experimental output in folder "OxfordMOPS_EXP/exp_d1" (where it seams to have been 0.54)."*

We thank the reviewer for this comment. Yes this is correct, the file "OxfordMOPS_exp_d1.json" does request DFO-LS to initiate from a detmartin value of 0.5. However, as seen in the Supplement/OxfordMOPS_EXP/exp_d1 files (prog_exp_d1.txt and dfo001/parameters_input.txt), DFO-LS actually initiates from 0.54. This is because DFO-LS cannot start too close to a parameter bound (determined by the DFO-LS setting "rhobeg"), and therefore shifts the starting values away from boundaries if necessary. DFO-LS always prints a warning when it does this, so the user is always notified, and OptClim ensures the parameter input file (e.g. dfo001/parameters_input.txt) contains the shifted parameter values.

This is fully reproducible. If OptClim were to re-run DFO-LS using "OxfordMOPS_exp_d1.json" exactly as it is, DFO-LS will always shift detmartin from 0.5 to 0.54, and the results will always be as in Supplement/OxfordMOPS_EXP/exp_d1.

Reviewer 2

1. *"I should have been clearer in my first review point 38:*
   *Eq. (30) + many lines: "fglobal". Usually non-variable subscripts are typeset upright. This comment should have mentioned all non-variable subscript and superscript. E.g., I would replace f^{Base}_\textrm{T} and r^{Base}_{qj} with f^\textrm{base}_\textrm{T} and r^\textrm{base}_{qj} (i.e., fBase and rBase with fbase and rbase) where I also avoided having a capital "B" for consistency. The authors should check the entire manuscript for any non-variable subscript/superscript and correct them."*

   We thank the reviewer for this comment. We have done as suggested.

2. *"As per GMD's guidelines (https://www.geoscientific-model-development.net/submission.html#math) vectors such as x, which I guess was LaTeX'd from \mathbf{x} (first appearance l. 117) should be typeset "in boldface italics", i.e., x. This is easily done using the \vec{x} command provided by the Copernicus LaTeX template."*

   We thank the reviewer for this comment. We have done as suggested.

3. *"Units are missing in almost every figure and should be added."*

We thank the reviewer for this comment. Units were missing from 4/13 of the figures (Figs 4, 9, 11, 13), but have now been corrected. The misfit shown in other figures is unit less.

4. *"l. 178: Eq. (2): the sums should not start at i=1. They should be written as \sum_{i\in j} instead of \sum_{i=1}^{i\in j} ($\Sigma i \in j$ instead of $\Sigma i=1i \in j$)."*

We thank the reviewer for this comment. We have done as suggested.

5. *"l. 132: Mathematical symbol D should be in italics, i.e., D. (Use "$D$".)"*

We thank the reviewer for this comment. We have done as suggested in Equation 1 and Line 123.

6. *"l. 168: "the length of the vector r" should be spelled out for clarity with maybe something like "d, the number of ri terms". (Also note that otherwise the vector r, which should be boldface italic, is not even defined.)"*

We thank the reviewer for this comment. Line 154 has been corrected as suggested, from

"There is no maximum suggested length of the vector $\boldsymbol{r}$, therefore …"

to

"There is no maximum suggested value for $d$, the number of $r_i$ terms, therefore …"

7. *"Table 1: In retrospect, my suggestions for experiment names were not great. For readability, I think it might be better to have shorter names and avoid underscores. What about:*
*"CTL" for the CMA-ES run (the control run),*
*"SMOOTH$_1$" for the "D_smooth_1 run" and so on,*
*"NOISY$_1$" for the "D_noise_rand1" run and so on, and*
*"SPARSE$_1$" for the "D_smooth_sparse_1" run and so on?"*

We thank the reviewer for this comment. Yes, we also agree that determining the best naming convention is difficult, as there is a balance to be found between containing the necessary information to describe each experiment, and keeping them short enough for good readability. It might be best to keep in the information as to which experiment is using which optimiser. Therefore, possibly a mix of the reviewer's previous suggested naming convention and the most recent suggestion above:

"C_SMOOTH" for the CMA-ES run (the control run),

"D_SMOOTH$_1$" for the "D_smooth_1 run" and so on,
"D_NOISY$_1$" for the "D_noise_rand1" run and so on, and
"D_SPARSE$_1$" for the "D_smooth_sparse_1" run and so on.

We realise this doesn't fully eliminate the underscores, but it is still an improvement to the 3 underscores of D_smooth_sparse_1 etc., and allows us to include the optimisation algorithm information. We have revised to now use this naming convention, and we hope the reviewer deems them suitable.

8. *"l. 221: Use the \times symbol rather than the letter "x"."*

We thank the reviewer for this comment. We have done as suggested.

9. We thank the reviewer for the comments below.

   *9A. "Fig. 4:*
   *Maybe a line for the target value could be added in the background? (and a +-5% band?) "*

   Thank you, this was left out in error and has been fixed.

   *9B. "Maybe show the 10 CTL (C_smooth) starting points as tiny dots?"*

   Thank you, this has been done as suggested, and the caption updated to explain this.

   *9C. "There is a lot of unused vertical space in each panel. Maybe the y-axis limits can be tightened a bit? E.g., Fig. 4c shows maximum K_PHY values of about 0.2, but the y-axis goes up to 0.5."*

   Thank you. The y-axes are fixed to the parameter bounds, within which DFO-LS was allowed to search, and allows the reader to understand where the starting and optimised parameter values fell within the bounded parameter space. Also, by adding the starting CMA-ES locations as suggested by the reviewer above, there is now less unused vertical space in each panel. For these two reasons the y-axes limits have been left unchanged, and we hope the reviewer agrees with this decision.

   *9D. "The legend could be simplified to only say that circles are starting points and crosses are optimized values? (Maybe use black for the legend and then give a different color than black for the smooth values."*

   Thank you, this has been done as suggested.

   *9E. "Speaking of color, a color-blind-friendly palette could be used here instead of plain black, red, and blue (e.g., colorbrewer's qualitative colors (https://colorbrewer2.org/#type=qualitative&scheme=Dark2&n=3), but there are many others!)"*

Thank you, this has been done as suggested in Figures 4 and 5. The colours in all other figures remain unchanged because the various marker styles ensure they are colour-blind friendly.

*9F. "The legend could be placed at the bottom rather than in the middle to avoid visually breaking the x-axis alignments of top and bottom panels."*

Thank you, this has been done as suggested.

10. *"Fig. 5: This is a key figure that was added in response to the 1st round of review to replace the now Table C. Yet, the main message — that DFO-LS requires much less evaluations than CMA-ES — is now obfuscated by the use of different scales and 2 y-axes. Better to show both on the same scale and let the visual speak for itself! The broken-axis suggestion (from the 1st review round) was not used, although it would make this much clearer in my opinion. Here is what I had in mind, e.g., for Fig. 5a (The red dashed line shows the imposed limit on evaluations for DFO-LS runs.): I understand MATLAB is not suited for broken-axis plots, so to be helpful I have provided below the python code that produces the broken-axis plot shown above. This code can easily be used as a template to reproduce each panel in Fig. 5. …"*

We thank the reviewer for this comment, and for the time they spent to modify the Python code they have kindly provided. We have used this code to create each panel of Figure 5.